# A distinct class of pan-cancer susceptibility genes revealed by an alternative polyadenylation transcriptome-wide association study

Hui Chen[1,10], Zeyang Wang[1,10], Lihai Gong[1], Qixuan Wang[1], Wenyan Chen[1], Jia Wang[2], Xuelian Ma[1], Ruofan Ding[1], Xing Li[1], Xudong Zou[1], Mireya Plass[3,4,5], Cheng Lian[6], Ting Ni[7], Gong-Hong Wei[6,8], Wei Li[9] ✉, Lin Deng[2] ✉ & Lei Li[1] ✉

Alternative polyadenylation plays an important role in cancer initiation and progression; however, current transcriptome-wide association studies mostly ignore alternative polyadenylation when identifying putative cancer susceptibility genes. Here, we perform a pan-cancer 3′ untranslated region alternative polyadenylation transcriptome-wide association analysis by integrating 55 well-powered ($n > 50,000$) genome-wide association studies datasets across 22 major cancer types with alternative polyadenylation quantification from 23,955 RNA sequencing samples across 7,574 individuals. We find that genetic variants associated with alternative polyadenylation are co-localized with 28.57% of cancer loci and contribute a significant portion of cancer heritability. We further identify 642 significant cancer susceptibility genes predicted to modulate cancer risk via alternative polyadenylation, 62.46% of which have been overlooked by traditional expression- and splicing- studies. As proof of principle validation, we show that alternative alleles facilitate 3′ untranslated region lengthening of *CRLS1* gene leading to increased protein abundance and promoted proliferation of breast cancer cells. Together, our study highlights the significant role of alternative polyadenylation in discovering new cancer susceptibility genes and provides a strong foundational framework for enhancing our understanding of the etiology underlying human cancers.

Genome-wide association studies (GWAS) have identified hundreds of single-nucleotide polymorphisms (SNPs) associated with increased risk of major human cancers[1,2], including breast[3], prostate[4], colorectal[5], and ovarian cancer[6]. In prostate cancer, for example, a highly heritable disease with 58% risk due to genetic factors, over 140 risk variants have been identified, explaining approximately one-third of familial risk for the disease[7]. Improving our understanding of inherited cancer-risk associated SNPs could provide new opportunities to elucidate the mechanisms of tumorigenesis. However, more than 90% of these variants are mapped to noncoding regions in the human genome[8], posing a significant challenge for their functional interpretation in regard to disease development, progression, and response to therapy.

Molecular quantitative trait locus (xQTL) analysis is a crucial step towards better understanding the effects of noncoding genetic variants on genes, pathways, and mechanisms of action, serving as an essential intermediate link between genotype and disease

phenotype[9–11]. Many molecular phenotypes derived from RNA sequencing (RNA-seq), such as gene expression and alternative splicing, have been used to discover disease-risk genes in population-scale xQTL studies[10,12,13]. Those genetic variants showing strong associations with the aforementioned molecular phenotypes are referred to as expression QTL (eQTL) or splicing QTL (sQTL). Such xQTLs can be highly informative; however, the effects of the numerous disease-associated noncoding variants remain unexplained[9,14].

Alternative polyadenylation (APA) has emerged as a new paradigm of post-transcriptional regulation for human genes[15–17]. That is, by employing different poly(A) sites, genes can either shorten or extend 3′ untranslated regions (UTRs) containing cis-regulatory elements, such as binding sites for microRNAs or RNA-binding proteins (RBPs)[18]. APA can affect target gene translation, as well as localization and protein–protein interactions of its gene product, independent of mRNA expression level or splicing[15,19]. Consequently, the diverse landscape of polyadenylation can significantly impact both normal development and disease progression[20,21]. In particular, individual genetic variants associated with APA have been linked to several cancers. For example, rs78378222 in the 3′UTR of *TP53* alters the canonical polyadenylation signal from AATAAA to AATACA; this impairs 3′-end processing of *TP53* mRNA, altering susceptibility to multiple cancers, including cutaneous basal cell carcinoma, prostate cancer, glioma, and colorectal adenoma[22]. In our previous study, we described the first atlas of genetic variants associated with APA (3′aQTL)[23], which highlighted that approximately 16.1% of GWAS loci colocalized with 3′ aQTL. Yet, this study did not include cancer GWAS loci, and therefore, the prevalence and functions of SNPs associated with APA for major human cancer types remain largely unknown.

In this study, we performed the first large-scale and systematic analysis assessing APA-mediated genetic effects on 22 cancer types in 49 human tissues from Genotype-Tissue Expression (GTEx) project[24] and 18 tumor tissues from The Cancer Genome Atlas (TCGA)[25] (Fig. 1a). APA transcriptome-wide association studies (TWAS) and identified 642 cancer susceptibility genes predicted to modulate cancer risk via APA, 62.46% of which are independent of gene expression and splicing. Furthermore, through a combination of genetic association analyses and experimental approaches, we validated APA-mediated risk genes linked to breast cancer, demonstrating that the alternative alleles of the 3′aQTL variant regulated 3′UTR usage of *CRLS1* leading to elevated protein abundance and thereby increasing the risk of breast cancer. Lastly, we have constructed a publicly available database with a user-friendly hub (http://bioinfo.szbl.ac.cn/TCGD/index.php) for use by the research community.

## Results

### Atlas of well-powered cancer GWAS summary statistics across 22 cancer types

To comprehensively characterize the genetic effects of APA on human cancers, we employed our Dapars v2.0 algorithm[23,26] to identify dynamic 3′UTR APA events from a large dataset of 23,955 genotype-matched RNA-seq samples obtained from 49 human tissues in the GTEx project and 18 tumor tissues in the TCGA dataset. We further used Matrix eQTL[27] to identify common genetic variants associated with differential usage of 3′UTR in each tissue or cancer type (see Materials and Methods) (Fig. 1a). We first compiled a large collection of 438 GWAS summary statistics from manually curated published studies and public cohorts, including the National Human Genome Research Institute–European Bioinformatics Institute (NHGRI–EBI) GWAS Catalog (release 2021/01)[28], the UK Biobank release 2 cohort (UKB2; release 2018/03)[29], the Japanese Encyclopedia of Genetic associations by Riken (JENGER)[30], and the FinnGen consortium (release 2021/05)[31]. After quality control and removing potential confounding factors such as duplicate patients, summary statistics from 55 reasonably powered GWASs ($N > 50,000$) across 22 cancer types were

retained for our analysis (Figs. 1b, S1–S3, and Supplementary Data 1). The median sample size across these studies was 194,153 individuals. We then extracted all lead SNPs, which represent the most significant cancer risk variants within a 1 Mb range. Our observation revealed that these lead SNPs tend to exhibit large effect sizes ($P = 1.07 \times 10^{-39}$, Wilcoxon rank-sum test, Fig. 1c), while their minor allele frequency (MAF) is evenly distributed (Fig. S4a) in comparison to the entire genome. We further performed functional annotations to determine the positional distribution of lead SNPs. 95.66% of GWAS lead SNPs were in noncoding regions, with 14.78% of them located in 3′UTR and downstream regions (20 kb, Fig. 1d), where lead SNPs in 3′UTR regions showed a strong enrichment against the entire genome (fold-enrichment = 1.62, $P = 1.88 \times 10^{-3}$, Fig. 1e). For example, the prostate cancer lead SNP rs4245739 was identified in the 3′UTR *MDM4*[32] (Fig. S4b), which encodes a regulator of p53, and breast cancer lead SNP rs1386230 was also located in the 3′UTR of *FGF10*[33] (Fig. S4b), which encodes the fibroblast growth factor 10. The lead SNPs within 3′UTR regions have effect sizes comparable to those in other genomic regions ($P = 0.114$, Wilcoxon rank-sum test, Fig. S4c, d).

To elucidate the extent of shared genetic architecture across different cancer types, we employed linkage disequilibrium score regression (LDSC)[34] to calculate heritability estimates from cancer GWAS summary statistics. Our analysis revealed that heritability estimates vary significantly across various cancer types. Specifically, lymphoma exhibited the lowest average heritability score ($h^2 = 0.0017$) with a 95% confidence interval (CI): 0.0004–0.0030, while breast cancer has the highest average heritability score ($h^2 = 0.17$), with 95% CI: 0.1637–0.1835 (Supplementary Data 2). These estimations align well with findings from previous individual studies[5,35]. To further explore the relationships between different cancer types, we conducted pairwise genetic correlation ($r_g$) analysis. Our result indicated that 22.09% of cancer pairs exhibited a significant correlation (Bonferroni-corrected threshold of $P = 0.05$). Interestingly, we observed that biologically related cancers tended to cluster together, exemplified by the clustering of basal cell carcinoma with melanoma, and breast with Female genital organs (Fig. S5). We further compared the genetic correlation within- and between- cancer types and found significantly higher genetic correlations within than between cancer types (average $|r_g| = 0.83$ and $|r_g| = 0.58$ respectively, $P = 0.0004$, Wilcoxon's test, Fig. S6a). Additionally, we explored the genetic correlation between different populations for each cancer type. Our findings revealed only weak genetic differences between within-European and within-Asian samples ($P = 0.5$, Wilcoxon rank-sum test, Figs. S6b, 1f, g). Taken together, these data underscore the importance of exploring genetic variants in 3′UTR and downstream regions, which constitute 14.78% of cancer GWAS SNPs, in the context of cancer risk. Moreover, our findings demonstrate that genetic factors contribute to the risk of certain cancer types while also exhibiting some degree of shared genetic components between biologically related cancers.

### 3′aQTL explains a significant proportion of cancer heritability

To assess the role of APA regulatory variants in human cancer and to evaluate the enrichment of 3′aQTLs within cancer GWAS loci, we applied a functional genome-wide association study (*fgwas*)[36] and observed 3′aQTLs are enriched within 35.88% of significant tissue–cancer pairs (Supplementary Data 3). When further compared with eQTLs, we found that 3′aQTLs had a large effect than eQTLs for 18.82% of tissue–cancer pairs, which include multiple relevant tissue–cancer pairs, such as the whole blood for prostate cancer (3′ aQTLs Enrichment = 1.68, 95%CI: 0.66–2.54, Fig. 2a) and adipose subcutaneous tissue for breast cancer (3′aQTLs Enrichment = 2.16, 95%CI: 1.07–2.87, Fig. 2b). Quantile–quantile plots (QQ-Plots) of GWAS *P* values for xQTLs and genome-wide SNPs further validate these results (Fig. 2c, d).

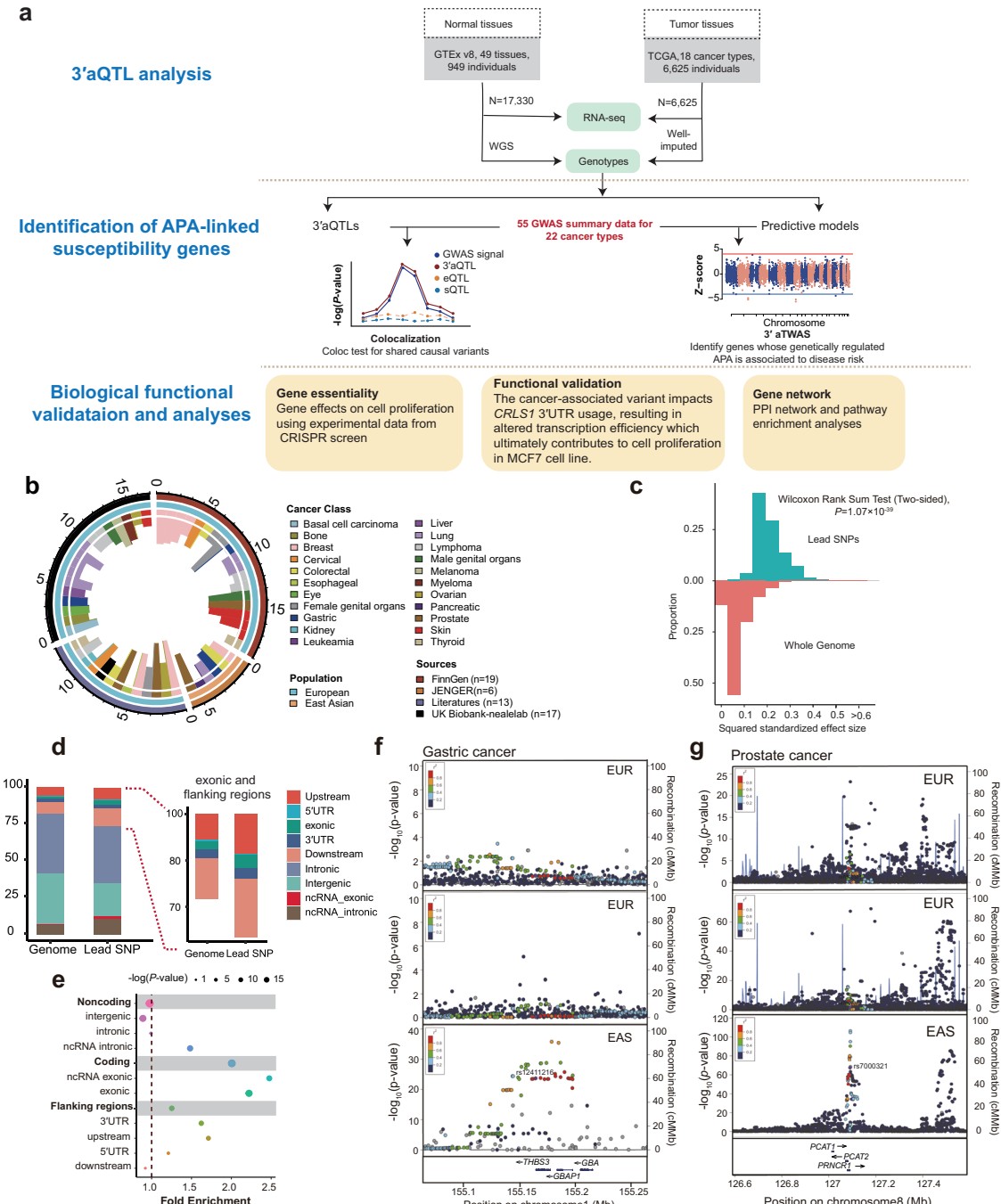

**Fig. 1 | Overview of study and collected cancer-related genome-wide association study (GWAS) summary statistics. a** Workflow of the entire study. Integration of RNA-seq and matched genotype data from GTEx and TCGA as reference panel for 3′aQTL analysis and 3′aTWAS modeling. We then performed 3′aTWAS analysis to identify APA-linked susceptibility genes using cancer GWAS summary statistics and 3′aTWAS models. The biological functions of APA-linked susceptibility genes were further validated by data analysis and experimental approach. **b** Atlas depicting genetic influences from 55 GWAS samples across 22 human cancer types. Inner histogram represents the log-transformed distribution of lead SNPs from each GWAS summary. Second circle highlights the different cancer types indicated by different colors. Third circle represents the different populations, including European and Asian people, indicated by blue and brown colors. Outer circle highlights the data sources in distinct colors. **c** Proportional histogram of squared standardized effect sizes for the lead SNPs, with minor allele frequency (MAF) > 0.01. **d** Distribution of functional consequences of lead SNPs versus all the SNPs in the genome. Different color represents the genomic annotation. The right panel is a zoom-in of annotation for the exonic and flanking regions (upstream, 5′ UTR, 3′UTR, and downstream). The strongest enrichment was seen in exonic ($E$ = 2.22) and followed by flanking regions ($E$ = 1.25) compared to the entire genome, while intronic and intergenic regions ($E$ = 0.98) were depleted. $E$, fold enrichment (proportion of SNPs with a certain annotation divided by the proportion of SNPs with the same annotation in the whole genome). **e** Enrichment analysis for positional distribution of SNPs with functional consequences. Each dot represents the fold enrichment of the proportion of lead SNPs with a certain annotation divided by the proportion of SNPs with the same annotation in the whole genome (background). The dot sizes represent the significance of the $P$-value, calculated from Fisher's exact test (two-sided). LocusZoom plot showing the significant loci associated with gastric (**f**) and prostate cancer (**g**) in Europeans and Asians; for example, rs12411316 in gastric cancer is only significant in the Asian population. EAS East Asian, EUR European.

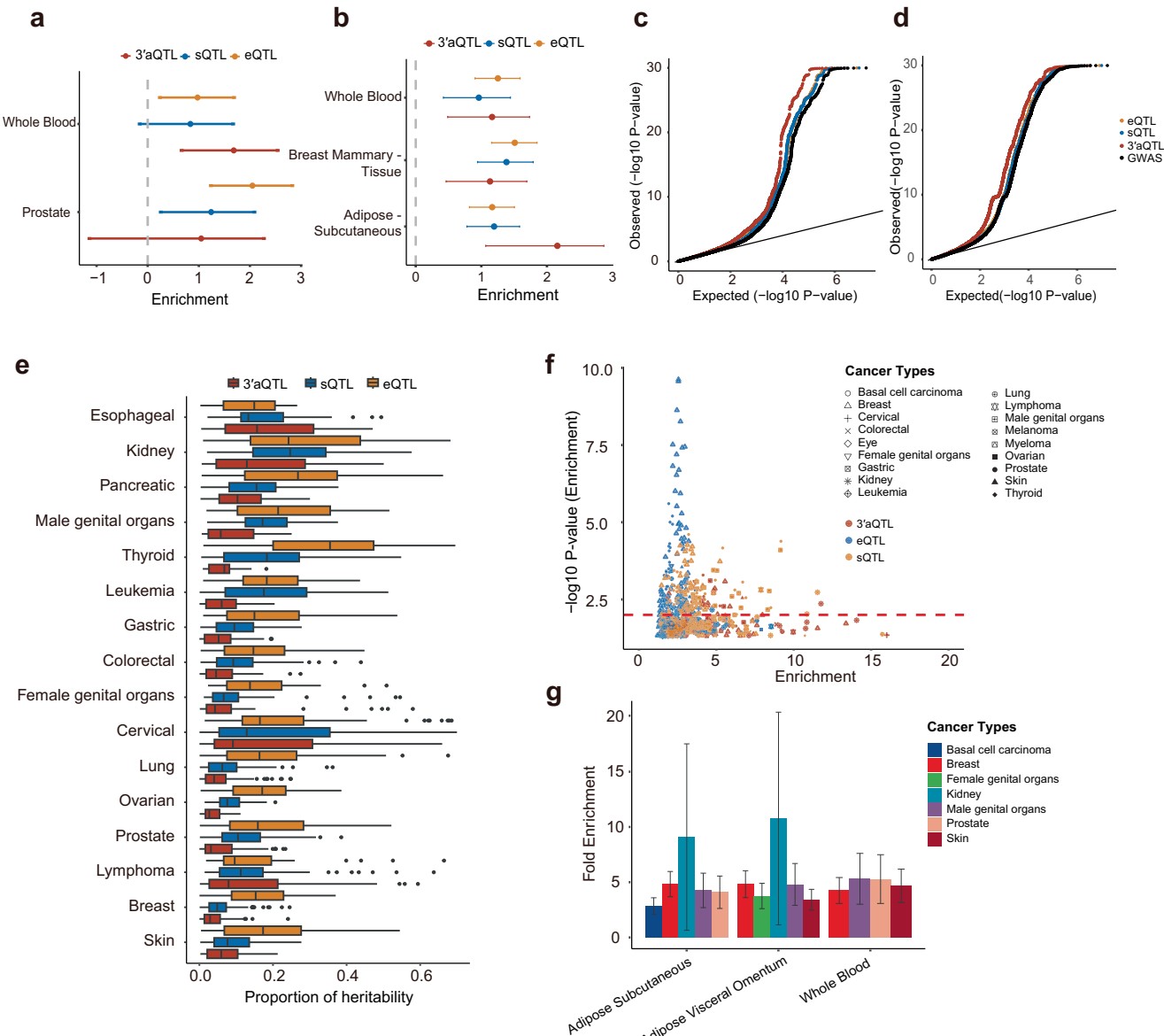

**Fig. 2 | Integrated analysis of 3'aQTLs enrichment for heritability of human cancer traits.** Enrichment of 3'aQTLs compared with eQTLs or sQTLs in relevant tissues for **a** prostate and **b** breast cancer. The effect sizes were estimated using functional genome-wide association analysis, which quantifies the enrichment of xQTLs in trait-associated variants. The estimated lower and upper bound 95% confidence intervals (CIs) for the effect sizes are also shown. Whole Blood ($n = 670$); Prostate ($n = 221$); Breast mammary tissue ($n = 396$); Adipose Subcutaneous ($n = 581$). Quantile–quantile plots of GWAS *P*-values for prostate cancer (**c**) and (**d**) breast cancer; 3'aQTLs, eQTLs, and sQTLs are compared to genome-wide SNPs. GWAS SNPs were binarily annotated using 3'aQTLs, eQTLs, and sQTLs, with a *P*-value < 0.05. **e** Partitioned heritability is determined by calculating the ratio of phenotypic heritability (*x*-axis) attributable to expression quantitative trait loci (eQTLs), splicing QTLs (sQTLs), and alternative polyadenylation QTLs (3' aQTLs) relative to aggregate SNPs for different cancer types (*y*-axis). The center

lines within the box plot signify the median values, while the boxes encompass the interquartile range (IQR) from the 25th to the 75th percentile. The whiskers extend up to 1.5 times the IQR (bottom) and the outliers are shown as separate dots. $n = 49$ tissues were examined. **f** Summary of GWAS heritability enrichment for cancer traits (with the most significant sample and case number) on the baseline linkage disequilibrium (LD) model. The dashed line shows the significant threshold at a $P < 0.05$. **g** Heritability enrichment in adipose-related and whole-blood tissues for multiple cancer types. Heritability enrichment is defined as a ratio of the proportion of heritability explained by the SNPs in query to the mean proportion observed in 1000 sets of control SNPs sampled repeatedly at random. Each column represents a point estimate with error bars representing the standard error. Adipose subcutaneous ($n = 581$); adipose visceral omentum ($n = 469$); whole blood ($n = 670$).

To further examine the proportion of 3'aQTLs associated with cancer heritability, we performed a partitioned heritability analysis using stratified (S)- LD score regression (LDSC)[37,38]. We observed that 3'aQTLs can contribute a median of 14.11% of heritability, compared to 18.96% for sQTLs and 33.07% for eQTLs (Fig. 2e, f). Expanding our S-LDSC analysis revealed that 3'aQTLs in whole blood and adipose tissues are enriched for associations with multiple cancer types, including cancers of the breast, male genital organs, prostate, skin, and

as well as basal cell carcinoma (Fig. 2g). Taken together, our findings strongly suggest that a significant proportion of cancer heritability could be attributed to 3'aQTLs.

## 3'aQTLs colocalize with cancer GWAS risk loci and are largely independent of gene expression and splicing QTLs

To systematically investigate whether cancer GWAS risk loci share the same causal variant with 3'aQTLs, we performed colocalization analysis

using *coloc*[39] (Supplementary Data 4). Our analysis revealed that 34 cancer GWAS traits across 17 cancer types colocalized with at least one type of molecular QTLs (Fig. 3a). Specially, we identified 766 eQTLs, 560 sQTLs, and 250 3'aQTLs that colocalized with cancer GWAS risk loci (Fig. 3b). In agreement with previous study[40], a large proportion of colocalization events was observed in long-noncoding RNAs across several cancer types, including colorectal cancer and lymphoma (Fig. 3b).

We then analyzed the proportion of cancer GWAS loci co-localized with these molecular QTLs and found that 3'aQTLs, eQTLs, and sQTLs co-localized with a median of 28.57%, 44.74%, and 35.36% of GWAS risk loci, respectively (Fig. 3c–e). Further analysis of the 3'aQTL-colocalized genes (3'aGenes) relative to eQTL and sQTL colocalized genes (eGenes and sGenes, respectively) revealed that, on average, 53.1% 3'aGenes were not associated with matched eQTLs or sQTLs

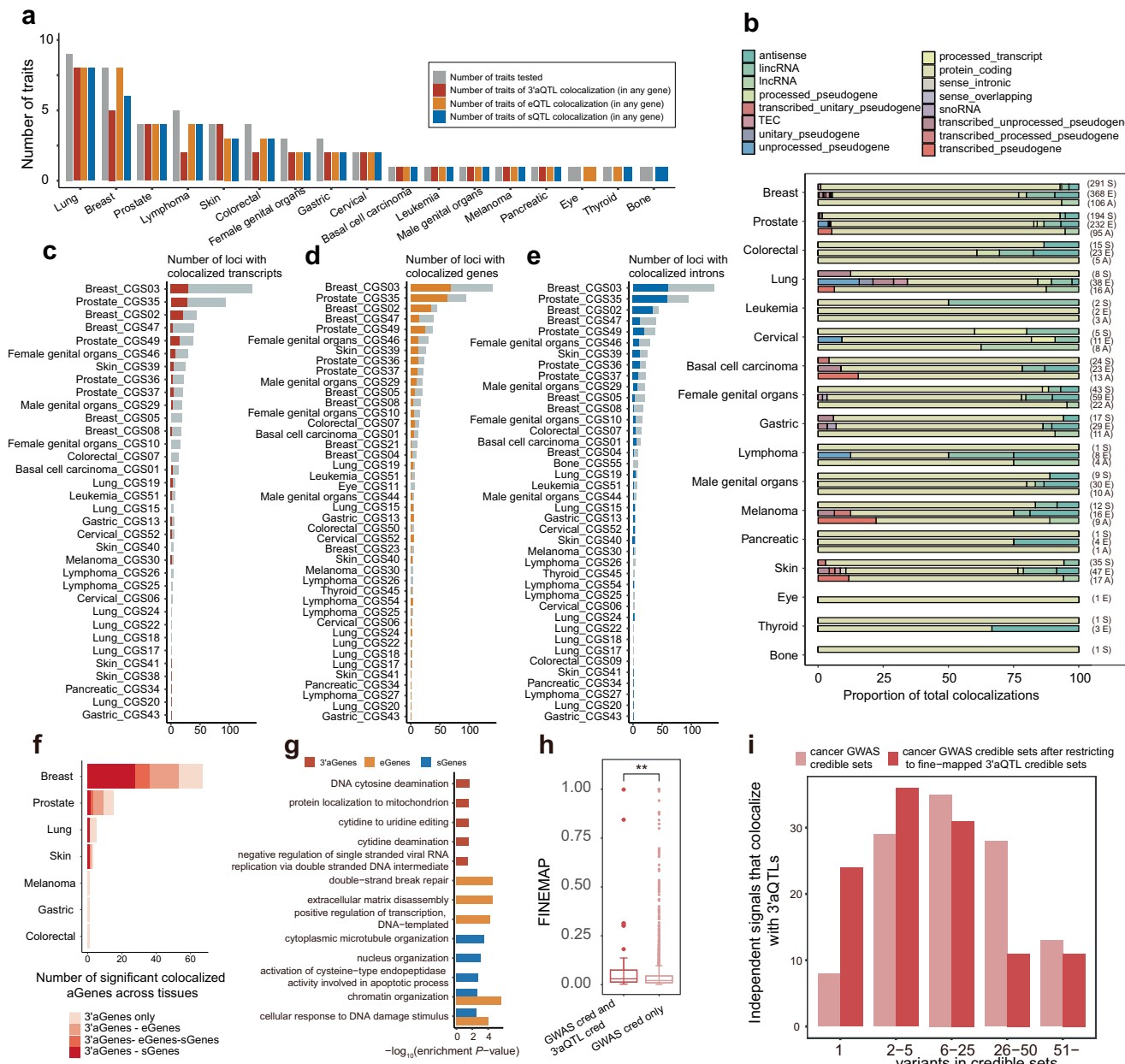

**Fig. 3 | Colocalization identifies cancer trait-associated genes. a** The number of specific traits for different cancer types that colocalize with 3'aQTLs, sQTLs, and eQTLs, indicated by the red, orange, and blue bars, respectively. **b** The frequency of significant colocalization events contributed by each gene type collapsed across tissues. GWAS data are grouped by general cancer categories on the *y*-axis. A missing bar indicates no colocalization for the trait category and QTL type. The numbers at the right of each bar indicate the total number of significant colocalization events (E, eQTL; S, sQTL; A, 3'aQTL). **c**–**e** The proportion of loci with a GWAS-significant variant that colocalizes with at least one gene expression (**c**), splicing (**d**), or APA (**e**) event. Across traits, a median of 44.74%, 35.36%, and 28.57% of GWAS loci colocalize with eQTLs, sQTLs, and 3'aQTLs, respectively. **f** Bar plot shows that 53.1% of 3'aQTL-colocalized cancer risk loci are specific for a 3'aQTL without a

corresponding eQTL or sQTL. **g** Pathway analysis of 3'aGenes, sGenes, and eGenes for pan-cancer. The bar represents the log-transformed enrichment *P*-value for each pathway. **h** Distribution of cancer GWAS causal posterior probabilities for all fine-mapped GWAS variants (95% credible set in GWAS, GWAS_cred) that are also fine-mapped 3'aQTL variants (3'aQTL_cred) *vs.* GWAS_cred variants only. Mann–Whitney (two-sided), $P = 2.22 \times 10^{-3}$. The center lines within the box plot signify the median values, while the boxes encompass the interquartile range (IQR) from the 25th to the 75th percentile and the outliers are shown as separate dots. **i** Integration of cancer-credible GWAS variants with credible sets from colocalizing 3'aQTLs increases fine-mapping resolution. The bar plot shows the number of independent loci identified as candidate causal variants before and after restricting for QTL variants.

among the analyzed tissue–cancer pairs (Fig. 3f). For example, we identified 3′aQTLs in the caspase 8 (*CASP8*) gene that strongly colocalized with GWAS risk loci across multiple cancers, including skin cancer (PP4 = 0.961), breast cancer (PP4 = 0.973), and basal cell carcinoma (PP4 = 0.915), but not with eQTLs nor sQTLs. Pathway analysis further revealed that 3′aGenes were enriched in several distinct pathways including protein localization to the mitochondrion, which contributes to the regulation of malignant transformation and tumor progression[41]. In contrast, eGenes and sGenes were enriched in pathways related to DNA damage repair, chromatin organization, and DNA demethylation processes, implying these genes may be more involved in processes related to maintaining genomic stability and regulating chromatin structure (Fig. 3g). Thus, our data suggest that 3′aQTLs colocalize with cancer GWAS risk loci, as observed in TCGA datasets (Fig. S7–10), and a large proportion of these 3′aQTLs occur independently of eQTLs and sQTLs.

To further explore whether 3′aQTLs were also enriched at causal cancer risk loci, we performed fine-mapping on colocalized 3′aQTLs and compared them with cancer GWAS risk loci within a 95% credible set. Our analysis demonstrated that GWAS credible set variants that were also 3′aQTL had a significantly higher posterior inclusion probability than those not shared with 3′aQTL (Mann–Whitney, $P = 2.22 \times 10^{-3}$; Fig. 3h). Furthermore, we observed that including 3′aQTLs strongly increased the genetic resolution of cancer GWAS credible sets, resulting in the identification of 53 risk loci with ≤5 putative causal variants compared to only 33 risk loci when 3′aQTLs are not considered (Fig. 3i). Furthermore, the sequence structure characteristics of colocalized 3′aGenes exhibited distinctions when compared to eGenes and sGenes. While 3′aGenes shared comparable 5′UTRs with eGenes and sGenes, they exhibited relatively shorter coding regions when compared with eGenes ($P = 1.75 \times 10^{-2}$) and sGenes ($P = 1.87 \times 10^{-3}$, Fig. S11) and much longer 3′UTR lengths when compared with eGenes ($P = 2.47 \times 10^{-15}$) and sGenes ($P = 8.82 \times 10^{-25}$, Fig. S11). Notably, a significantly higher prevalence of adenylate-uridylate-rich (AU-rich) elements proximal to poly(A) sites was observed in colocalized 3′aGenes in comparison to both eGenes ($P = 6.43 \times 10^{-9}$) and sGenes ($P = 3.04 \times 10^{-9}$), suggesting that 3′aGenes harbor an increased number of potentially regulatory elements and have higher possibility of enhanced post-transcriptional regulation (Fig. S12). Taken together, these results suggest that the identified 3′aQTLs likely contain cancer causal risk variants. Furthermore, our findings indicate that 3′aGenes are largely distinct from eGenes and sGenes across many cancer types.

## APA transcriptome-wide association analysis reveals novel cancer susceptibility genes

To systematically identify and prioritize candidate APA genes associated with human cancers, we performed a multi-tissues 3′UTR APA transcriptome-wide association study (3′aTWAS) using the data from 49 GTEx tissues and 18 TCGA tumor tissues. The goal of this analysis was to estimate the association between genetically predicted APA usage and cancer risk, utilizing transcriptome panels with our curated cancer GWAS summary statistics[42]. For each tissue, we employed FUSION[42] to estimate the heritability of 3′UTR APA usage explained by *cis*-SNPs located in the 3′UTR of each transcript using linear mixed-linear models (Top1, BLUP, LASSO, and Elastic Net). To ensure robustness, cross-validation was used to select the best-fitted model with optimal prediction accuracy for each gene. This analysis resulted in 110,501 tissue-specific prediction models (Fig. S13), encompassing 25,934 APA events. To evaluate the prediction accuracy, we calculated the correlation between predicted and observed 3′UTR usage and further normalized by heritability. In line with previous TWAS studies[42,43], the average in-sample prediction accuracy is 68.52%, indicating that most of the signal in genetically predicted 3′UTR APA usage level is captured by the fitted models (Fig. S14). Moreover, we observed a high correlation between the number of 3′aTWAS predictive models

and the sample sizes of the GTEx reference panels (Spearman correlation R = 0.83, $P = 1.46 \times 10^{-13}$).

We applied our prediction model to cancer GWAS summary statistics and identified 642 APA-linked susceptibility events (FDR < 0.05, Fig. 4a, b and Supplementary Data 5), 62.46% of which were overlooked by conventional gene and splicing TWAS analyses (Fig. 4b). Based on the prediction model from the relevant tissues and cancer types, we still identified 276 APA events (Fig. S15). Additionally, there are 47 genes that were identified both in the colocalization and 3′aTWAS analyses (Supplementary Data 6) and seven genes that were consistently identified in 3′aTWAS analyses for both GTEx and TCGA datasets (Supplementary Data 7). Interestingly, our 3′aTWAS identified multiple known cancer risk genes, such as small G protein signaling modulator 3 (*SGSM3*)[44], which is significantly associated with breast cancer in breast mammary tissue ($P_{3′aTWAS} = 2.6 \times 10^{-18}$, $P_{eTWAS} = 0.47$, $P_{sTWAS} = 0.40$). This finding suggests that 3′UTR APA usage of *SGSM3*, rather than the expression or the splicing of the *SGSM3* gene, mediates breast cancer risk.

We then analyzed the identified genes across cancer types and found that breast and prostate cancers have the greatest number of significant APA-linked susceptibility genes (Fig. 4b), which may be due to the high heritability of these two cancer types (Fig. S16b and Supplementary Data 2) and not attributed to sample size differences (Fig. S16a). Moreover, many of these genes are shared across multiple cancer types (Fig. 4c), such as the pan-cancer 3′aTWAS gene sorting nexin 17 (*SNX17*), which plays a role in signaling-receptor and phosphatidylinositol binding, and is associated with breast and lung cancer[45]. Similarly, the 3′aTWAS gene spermatogenesis-associated 33 (*SPATA33*), which encodes a mammalian germline mitophagy receptor, is simultaneously related to the risk of melanoma, basal cell carcinoma, and skin cancer. Three APA genes common to breast and skin cancer were also identified, including the known tumor suppressor gene *CASP8*, a central mediator of the extrinsic apoptosis and necroptosis pathways[46] (Fig. 4c).

To further assess the functional roles of putative APA-linked cancer susceptibility genes, we determined the effect of gene silencing on proliferation in cancer-relevant cell lines based on data from CRISPR-Cas9 gene essentiality screens[47] (Fig. 4e, f and Fig. S17). We utilized the CERES score to represent the gene essential levels, which corrects for the computational effects of copy number and depletion of gene-targeting guide RNAs. A lower CERES score indicates a high degree of gene essentiality in each cell line. Our results revealed that 27 APA-linked genes identified in the potentially relevant tissues demonstrated evidence for essential roles in cancer cell proliferation (CERES score<−0.5). Remarkably, eight genes exhibited similar or even higher levels of essentiality compared to the well-known oncogene *MYC*[48] (Supplementary Data 8), including *PHF5A*, *EIF2S2* in basal cell carcinoma, *RPS23*, *DYNC1I2*, *POLR3C* and *RPAIN* in breast cancer, *UTP4* in skin cancer, *PHF5A* in cancer of female genital organs and *CACTIN* in cancer of male genital organs. Moreover, we observed that the mean CERES scores of 3′aTWAS genes were significantly lower than the non-3′aTWAS genes ($P = 9.8 \times 10^{-10}$, Wilcoxon signed-rank test, Fig. 4d). Additionally, the mean CERES scores of 3′aTWAS genes were also significantly lower than eTWAS genes ($P = 6.28 \times 10^{-5}$) and sTWAS genes ($P = 6.91 \times 10^{-4}$, Fig. S18). This indicates that APA-linked susceptibility genes tend to have a greater impact on cancer cell proliferation. To further validate our findings, we examined another larger-scale drop-out data deep RNAi interrogation of viability effects in cancer (DRIVE)[49], and identified six 3′aTWAS genes that were also essential. These genes included *BPTF*, *PHF5A*, and *SPG7* in cancer of female genital organs, as well as *FGF10*, *NFIX*, and *SCAP* in breast cancer. Collectively, our APA transcriptome-wide association analysis successfully identified numerous known and novel susceptibility genes with functional implications in multiple cancer types.

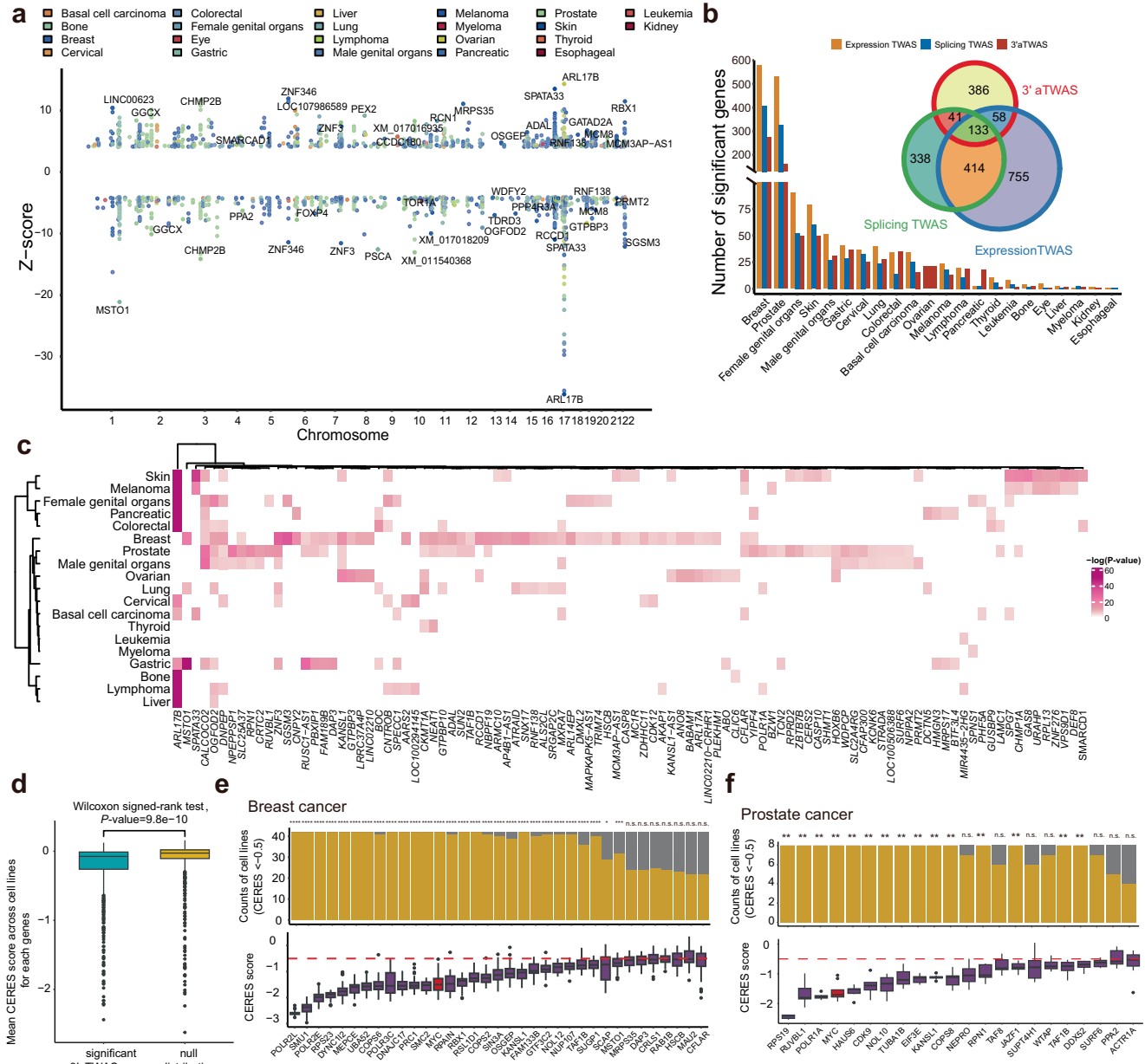

**Fig. 4 | APA Transcriptome-wide association study (TWAS) results. a** Manhattan plot of 3'aTWAS nominating the pan-cancer susceptibility genes. Significant 3' aTWAS associations at false-discovery rate (FDR) < 0.05 were shown. Colored points represent different cancer types. **b** Bar plots showing the number of significant genes detected by 3'aTWAS, with FDR < 0.05, for different cancers in the multiple tissues. Venn plot shows the intersection of significant cancer genes identified by 3'aTWAS (FDR < 0.05) with genes identified by expression and splicing TWAS. **c** Heatmap showing the 3'aTWAS genes shared across different cancer types. The color represents the log-transformed *P*-values for 3'aTWAS results. **d** Comparison of the mean CERES score across significant 3'aTWAS genes versus null distribution (randomly sampled in insignificant eTWAS, sTWAS, and 3'aTWAS genes). *P*-value was calculated from the Wilcoxon signed-rank test (two-sided), *n* = 720. Effect of cancer-susceptibility-associated APA-linked genes on cell

proliferation of **e** breast cancer (*n* = 45) and **f** prostate cancer-related (*n* = 8) cell lines. *MYC* is a known essential gene set as positive control based on experimental data from DepMap. CERES score to represent the gene essential levels, which corrects for the computational effects of copy number and depletion of gene-targeting guide RNAs. Red dashes denote the median CERES cutoff value of <−0.5, which indicates an essential role in cell proliferation. The significance of cell proliferation was tested for each gene based on the count of CERES values < −0.5 in a total of respective relevant cells using the Binomial test. **P < 0.01; *** P < 0.001; **** P < 0.0001; ns not significant. The center lines within the box plot (of **d**–**f**) signify the median values, while the boxes in each plot represent the first and third quartiles; the whiskers extend to the 1.5 times the IQR and the outliers are shown as separate dots.

### CRLS1 is a novel APA-mediated breast cancer susceptibility gene

Our colocalization and 3'aTWAS results identified multiple novel APA-linked cancer-susceptibility genes. Among these candidates, we focused on cardiolipin synthase 1 (*CRLS1*) for further experimental validation due to its identification in both colocalization and 3'aTWAS analysis from GTEx and TCGA data (Figs. 5a and S19a). Furthermore, *CRLS1* showed conditionally independent at the associated breast

cancer loci (Fig. 5b), implying that APA-mediated risk variants substantially explain the GWAS signal within this genomic region. To functionally assess the role of alternative alleles of 3'aQTL in tumor cellular phenotypes, we first observed a higher expression of the long 3'UTR isoforms with alternative alleles of 3'aQTL (rs2235816 G > A) (Figs. 5c, d and S19b, c). We validated this observation experimentally by performing 3' rapid amplification of complementary DNA ends

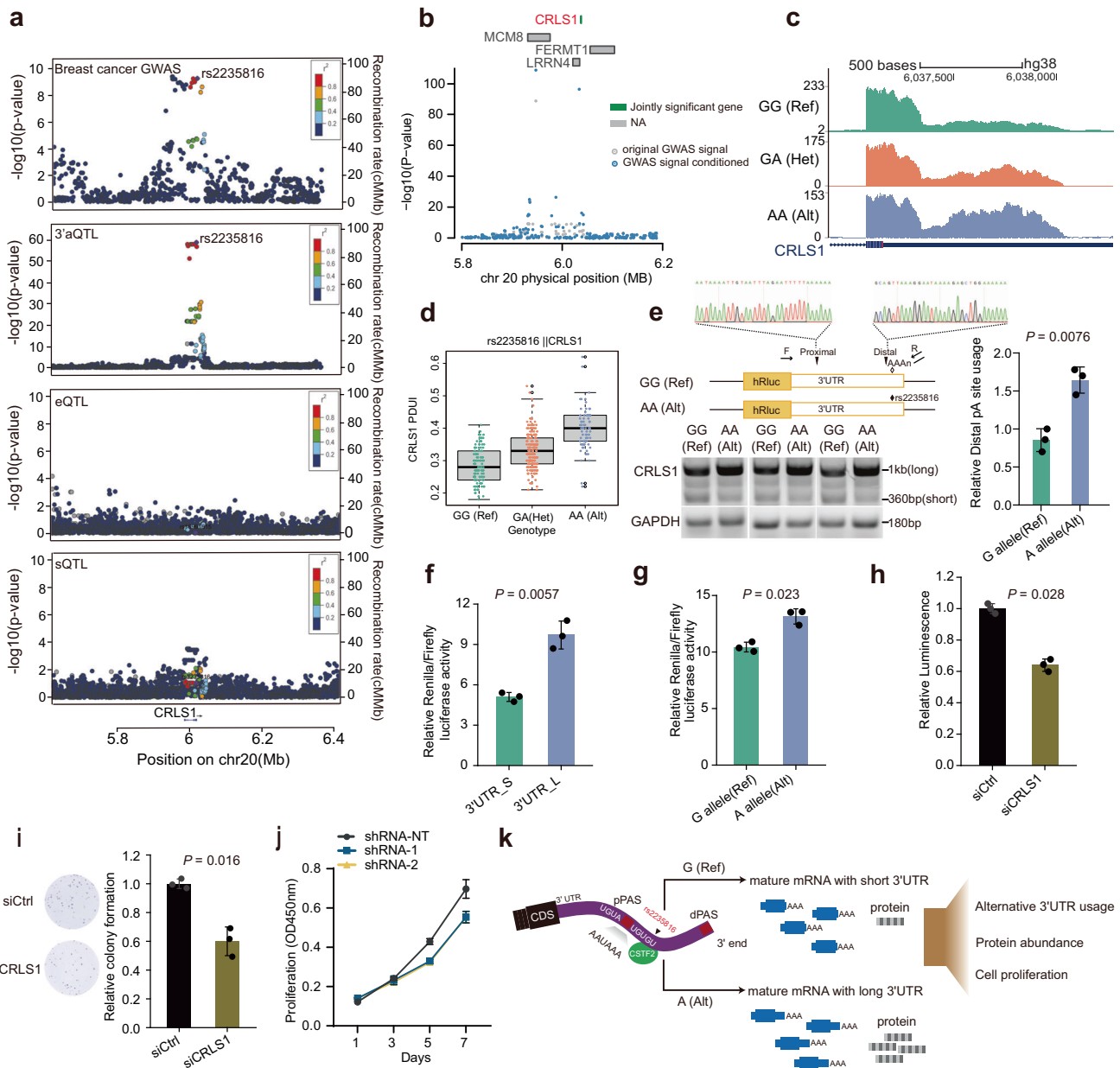

**Fig. 5 | APA-linked susceptibility genes in breast cancer. a** LocusZoom plot of breast cancer GWAS SNPs, eQTLs, sQTLs, and 3'aQTLs at the *CRLS1* locus. SNPs are colored by LD (*r²*). **b** Region association plots for *CRLS1*. Breast cancer GWAS signal at the gene locus (gray) and the GWAS signal after removing the effects of target gene 3'UTR usage (black); results indicate that the association is largely explained by target gene 3'UTR usage. **c** Example of RNA-seq coverage plot for the *CRLS1* 3' UTR. **d** Box plot showing the distribution of the normalized PDUI values for each genotype, each dot in the box plot represents the PDUI value for one individual from the breast mammary tissue of GTEx project (*n* = 396). **e** Changes of 3'UTR usage in MCF-7 cells detected by 3' RACE. The 3' RACE strategy is shown in the upper panel. F and R indicate forward and reverse primers, respectively. The bar plot on the right shows the measured usage of dPAS. (*n* = 3; *P* = 0.0076). **f** Luciferase activity from a reporter system containing the short and long 3'UTR of *CRLS1* in MCF-7 cells (*n* = 3; *P* = 0.0057). **g** Luciferase activity from a reporter system containing reference and alternative variant of 3'aQTL located within 3'UTR region in MCF-7 cells (*n* = 3; *P* = 0.023). **h** Proliferation of MCF-7 cells treated with the indicated siRNAs (*n* = 3; *P* = 0.0028). **i** Colony formation assay for MCF-7 cells treated with the indicated siRNAs. Left panel: Colonies were formed in 12-well plates and imaged on day 8 after siRNA treatment. Images are representative results from three independent experiments. Right panel: Quantification of colony numbers from the left panel (*n* = 3; *P* = 0.016). **j** Cell proliferation of shRNA-mediated knockdown cells was analyzed on days 1, 3, 5, and 7 (*n* = 3). **k** Schematic depicts alternative alleles of 3'UTR variants mediated 3'UTR lengthening promotes cancer progression. For panels **e**–**j**, the data shown are the mean ± s.d. from three independent experiments. *P* values were calculated using the two-tailed, paired Student's t-test. Source data are provided as a Source Data file.

(RACE) in MCF-7 breast cancer cells (Fig. 5e). To further investigate the contribution of APA in regulating *CRLS1* protein abundance, we performed luciferase reporter assays and inserted the shorter 3'UTR and the longer 3'UTR with a mutated proximal polyadenylation signal into a dual-luciferase reporter system. The reporter containing the longer 3'UTR exhibited significantly higher luciferase activity than the construct containing the short 3'UTR (Fig. 5f). These data imply that the

extended 3'UTR promotes elevated protein levels of *CRLS1*. Furthermore, we assessed the impact of different alleles of 3'aQTL on protein abundance using luciferase reporter assays with the reporter gene containing the 3'UTR region. Notably, the reporter containing alternative alleles significantly increased the luciferase signals compared to the reference alleles (Fig. 5g). We identified that rs2235816 resides within the binding motif of *CSTF2* (cleavage stimulation factor 2)[50,51], a

known APA regulator that potentially modulates the biological effect of rs2235816 (Fig. S20)[52]. These findings suggest that alternative alleles of 3′aQTL, which mediated APA changes, could contribute to the upregulation of *CRLS1* protein abundance.

Next, we investigated the phenotypic consequences of *CRLS1* dysregulation to mimic the APA-mediated protein changes. We first silenced *CRLS1* expression using small-interfering (si)RNAs and analyzed its effects on cell proliferation. Knockdown of *CRLS1* significantly inhibits cell proliferation and decreases the colony formation compared to the non-targeting control (Figs. 5h, i and S21a). We also depleted *CRLS1* with two short hairpin RNAs and observed reduced proliferation (Figs. 5j and S21b). These experimental results are in line with our 3′aTWAS analysis result, which suggested that the usage of long 3′UTR of *CRLS1* increases breast cancer risk (Z-score = 6.10 in breast mammary tissue). We also selected three other top-ranked putative breast cancer APA-linked susceptibility genes-autocrine motility factor receptor (*AMFR*), RPA-interacting protein (*RPAIN*), autophagy related 10 (*ATG10*) (Fig. S22a–c) and observed consistent results that alterations in alternative 3′UTR of these genes could affect protein abundance (Fig. S22d–f) and cellular proliferation (Fig. S22g–l). Taken together, our findings identified several susceptibility APA genes and supported the conclusion that alternative alleles of 3′aQTL influence APA patterns of *CRLS1*, resulting in longer 3′UTR usage and elevated protein levels. Consequently, these changes may contribute to an increased risk of breast cancer (Fig. 5k).

### Known and novel cancer susceptibility APA genes form coherent functional pathways

To investigate whether our identified APA-linked susceptibility genes were connected to known cancer genes via the same network or pathway, we accessed its presence in publicly available cancer-related gene sets from the literatures[53,54], the Discovery of Oncogenes and tumoR suppressor genes using Genetic and Epigenetic features (DORGE)[55], the Cancer Gene Census (CGC)[56] and the Molecular Signatures Database (MSigDB)[57]. Notably, we identified 60 genes with well-established role in cancer biology, such as for breast cancer (*BAZ3A, DNAJA1, SMU1, CDK12, KANSL1, SLC4A7, L3MBTL3, MRTFA, NF1, VEZT, SIN3A, AKAP9, TAF1B, NFIX, MEPCE, CHD3, CASP8, TRIOBP*), for basal cell carcinoma (*EIF2S2, VMP1*), for cervical (*PAX8, CDK12, ERBB2, PNISR*), for colorectal (*SMARCD1, ARFGEF2, LTBP2, CNOT9, PREX1*), for female genital organs (*SRSF3, TKT, SH3PXD2A, VPS37B, HRAS, DNAJB12*), for lung cancer (*POLR1A, TBC1D2B*), for lymphoma cancer (*POLR1A*), for male genital organs (*KANSL1, USP36, CRKL*), for ovarian cancer (*KANSL1*), for prostate cancer (*SEC62, CCND3, SOD2, AXL, RPRD1, ZBTB7B, NUCKS1, POLR1A, WTAP, TAF1B, FLT3LG, BCL2L12, SMAD2, POGZ, RPS19, KANSL1, NCOA4, ZBTB16, CTBP2, EIF3E, GATA2, SETDB1, IGMBP2, RAPGEF3*) and skin cancer (*ZBTB7B, ANKRD11, KANSL1, RPRD2, CASP8, SMARCD1*).

Further, we used STRING[58] to measure protein–protein interaction (PPI) network connectivity between the products of genes identified in our study and known cancer-susceptibility proteins. Our results show that members of the 3′aTWAS-prioritized PPI network exhibited significant enrichment for key biological pathways (Fig. 6). These pathways included the intracellular protein transport ($P = 1.13 \times 10^{-2}$), a critical process involved in metabolic reprogramming during cancer progression and adaptation to new environment[59]. Furthermore, our analysis revealed strong enrichment of APA-linked susceptibility genes in pathways associated with apoptosis and necrosis, such as the execution phase of apoptosis ($P = 8.75 \times 10^{-4}$) and TRAIL signaling ($P = 6.07 \times 10^{-4}$), which are essential to tumorigenesis[60]. These results indicate that the novel candidate genes identified in our study are broadly involved in gene networks crucial for cancer growth. Importantly, our results demonstrate that our novel cancer-susceptibility genes form coherent functional pathways with known cancer genes like *TP53*

and *BRCA1/2*, suggesting that many of these novel genes are likely to have a functional role in cancer risk.

## Discussion

Dysregulation of APA has been frequently observed in human primary cancer[61] and cancer cell lines[62], suggesting its potential involvement in cancer pathogenesis. Although germline genetic variants in the 3′UTR region have been linked to transcriptome diversity and various human diseases[23], the regulatory mechanisms underlying specific APA events and functional genetic variants associated with cancer susceptibility genes have remained largely elusive. In this study, we conducted the first comprehensive investigation to understand how APA-mediated genetic variations contribute to human cancer risk. We demonstrate that 95.66% of GWAS lead SNPs are noncoding, and 14.78% were located in 3′UTR and downstream 20 kb regions. Furthermore, our findings provide additional evidence that functional common genetic variants linked to alternative 3′UTR usage can contribute to the cancer risk and may play a substantial role in cancer initiation. Interestingly, this mechanism implicates APA-mediated GWAS risk loci that are not conventionally associated with dysregulated gene expression or splicing. Through the application of 3′aTWAS to our well-powered GWAS summary statistics, we identified 642 APA-linked susceptibility genes across multiple tissues. Remarkably, a substantial portion of these genes (62.46%) were previously overlooked in traditional expression and splicing studies, thereby expanding the repertoire of cancer risk loci annotated with potentially new regulatory mechanisms. To validate the predicted effects of 3′aQTLs on 3′UTR usage, we experimentally validated the genetic regulation of *CRLS1* in breast cancer. We demonstrated that the cancer-associated noncoding variant impacts 3′UTR usage, resulting in altered protein abundance which ultimately contributes to cell proliferation. Further analysis revealed the enrichment of 3′aTWAS-identified genes in known cancer pathways and essential gene sets, further substantiating their biological relevance.

In particular, our investigations revealed a 3′aTWAS association for *CRLS1* in breast cancer, a finding that was validated through experimental evidence. The 3′ RACE result demonstrated a significant difference in APA usage between the two alleles of rs2235816. The alteration in 3′UTR usage may result from the 3′aQTL disrupting the binding sites of RNA-binding proteins such as *CSTF2* (Fig. S20). This would be in agreement with a previous report that demonstrated increased use of distal PAS upon *CSTF2* depletion[52]. The luciferase assay demonstrated a significant difference in translation efficiency between the two alleles of rs2235816. This may be due to *CRLS1* proteins are predominantly regulated by translation of the long 3′UTR isoform, which contains binding sites for the translation regulator. We found that the longer 3′UTR isoforms provided more frequent RBP binding sites (Fig. S23a, b), indicating it was more likely to be regulated by certain translational regulatory RBPs. We indeed observed the U-rich motif for *CPEB2*[63] on *CRLS1* 3′UTR region, which has been demonstrated that could increase the protein abundance[64]. The observations that longer 3′UTR isoforms could generate higher protein levels have also been found in several other studies[64–66]. For example, the long 3′UTR isoform of Uncoupling protein 1 (*UCP1*) contributes only 5–10% of the total *UCP1* mRNA level, but its deletion reduced *UCP1* expression by 50–60%. Cytoplasmic polyadenylation element- binding protein 2 (*CPEB2*) binds to the long 3′UTR isoform of the *UCP1*, which increases translation efficiency and protein abundance of *UCP1*[64]. Together our data suggest that the allele-specific activity of risk SNPs play a significant role in elucidating the functional impact of alternative alleles of 3′aQTL in mediating changes in 3′UTR usage, leading to elevated protein abundance that exerts a pivotal influence on cancer proliferation.

However, we acknowledge several caveats and suggest future research directions. First, while we focused on common germline variants in this study, the contribution of rare variants to cancer

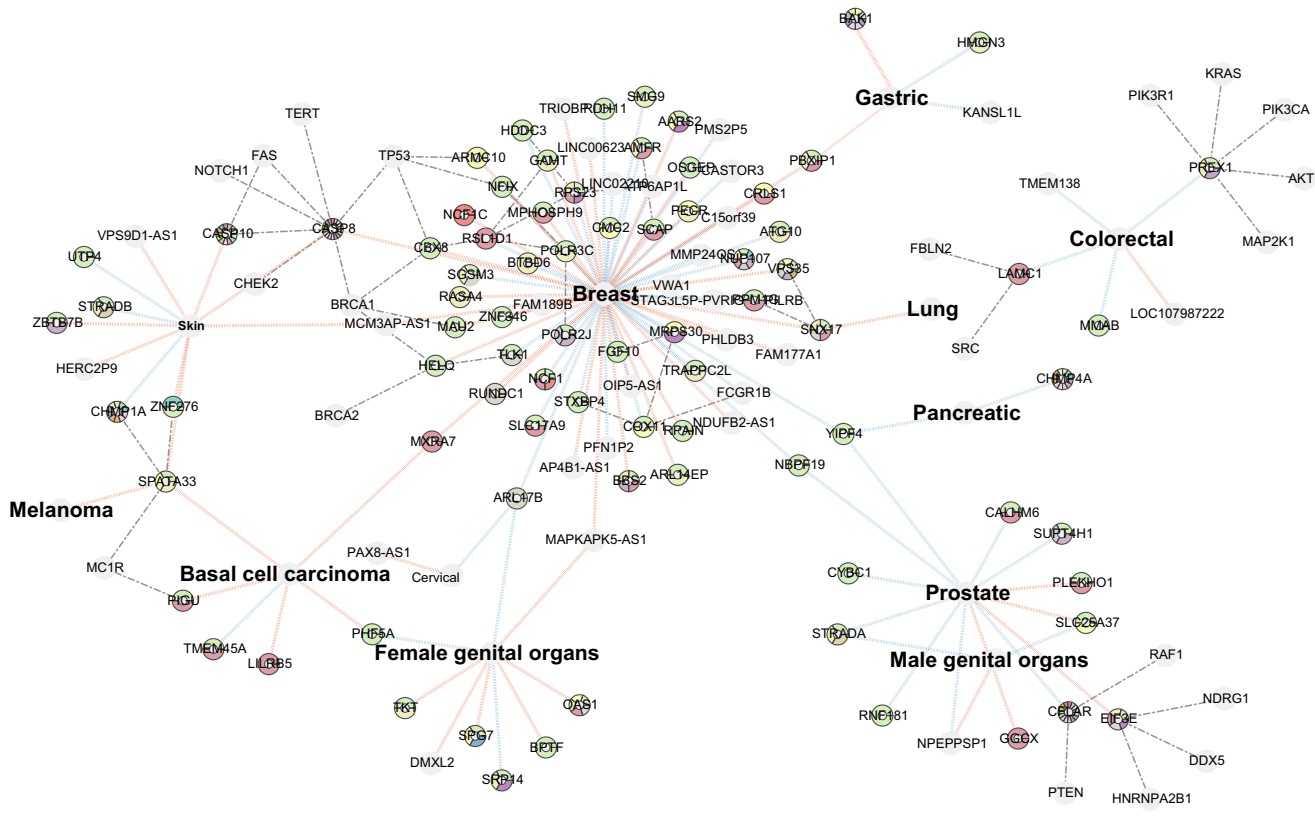

### Biological Process(Gene Oncology)

◯ GO:0006886~intracellular protein transport P=1.13E-02

### Molecular Function(Gene Oncology)

◯ GO:0005515~protein binding P=1.92E-02

◯ GO:0097153~cysteine-type endopeptidase activity involved in apoptotic process P=8.75E-04

### REACTOME PATHWAY

◯ R-HSA-75158~TRAIL signaling P=6.07E-04

◯ R-HSA-5218859~Regulated Necrosis P=2.85E-03

### Edge

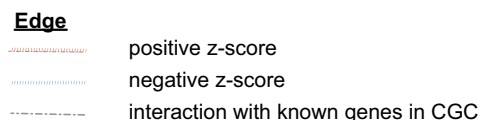

positive z-score
negative z-score
interaction with known genes in CGC

**Fig. 6 | Newly identified APA-linked cancer susceptibility genes are enriched in metabolic reprogramming and apoptosis-related pathways.** APA-linked genes are connected in protein–protein interaction (PPI) networks with known cancer-risk genes, such as *CASP8* in breast and skin cancer. Pathway enrichment analysis shows that 3′aTWAS genes are enriched in cancer-related pathways, including the metabolic reprogramming pathway like intracellular protein transport ($P = 1.13 \times 10^{-2}$), the apoptosis-related pathway ($P = 8.75 \times 10^{-4}$) and TRAIL signaling pathway

($P = 6.07 \times 10^{-4}$). Each node represents one APA-linked gene. Blue links between nodes represent a negative Z-score in 3′aTWAS, implying that usage of the short 3′ UTR increases cancer risk, whereas red lines represent positive z-scores, indicating that usage of the longer 3′UTR leads to increased cancer risk. Gray dashed line indicated the PPI interactions between APA genes and the known cancer genes in CGC.

susceptibility remains unknown and warrants further investigation. Second, our current study primarily focused on the overall risk for each cancer type, potentially overlooking cell-type-associated cancer susceptibility genes. Future investigations should consider molecular subtypes within cancers, such as breast cancer, to gain a more comprehensive understanding of the genetic landscape. Finally, we believe that our findings warrant the integration of other transcriptome-wide association study-like methods to explore regulatory mechanisms for susceptibility genes by incorporating information on *cis*-regulatory elements, such as RNA-binding proteins (RBPs) and microRNA.

In summary, our study presents multiple promising gene candidates for subsequent experimental follow-up, with a focus on investigating the molecular mechanisms underlying cancer progression and development. Further exploration of these genes holds great potential for providing new mechanistic insights into the pathology and genetics of various cancers. We have made the results and data of this study publicly available through our resource (http://bioinfo.szbl.ac.

cn/TCGD/index.php), which serves as an important asset for the research community to interpret the function of cancer risk loci in a wide variety of human cancers. Moreover, in-depth studies of APA-linked genes identified in our study will undoubtedly shed more light on the biological regulation and disease etiology of human cancers.

## Methods

### Collection and quality control of cancer GWAS summary statistics

We collected and integrated cancer GWAS summary statistics across 22 cancer types from four different sources: (1) GWAS summary statistics reported in the literature[3–6,67,68]; (2) UK Biobank GWAS[29]; (3) FinnGen[31] and (4) JENGER[30]. We removed the studies with less than 50,000 individuals and with potentially duplicated patients or controls. We also performed the quality control analysis of the remaining cancer GWAS summary statistics data. We first calculate the lambda value to estimate the *P*-values inflations. Then, we evaluate the quality

by examining the distribution of observed *P* values with the expected distribution using QQ-plot. Later *P*–*Z* plot was used to examine the normality of the distribution of Z-scores (Figs. S2 and S3). As a result, 55 well-powered cancer GWASs were used in this study. To harmonize the cancer GWAS summary statistics due to different genome versions, CrossMap[69] was used to convert the GWAS coordinates to the human genome assembly hg19/GRCh37.

## Annotation and characterization of lead SNPs and estimated standardized effect sizes

Lead SNP refers to the statistically most significant variant for cancer risk loci. Briefly, we first performed double clumping by PLINK (v.1.90)[70], which reports independent significant SNPs with *P*-value < 5 × 10$^{-8}$ and are independent at $r^2 < 0.6$. We further performed second clumping based on these independent significant SNPs, and lead SNPs are defined if they are independent of each other at $r^2 < 0.1$. Circos (v.0.69–9)[71] was used to visualize the distribution of lead SNPs for each cancer GWAS.

Further, the distribution of functional consequences of the lead SNPs, which were annotated with dbSNP147 databases (avsnp 147, build hg19) using ANNOVAR[72], was compared with all SNPs in the genome. The flag "–neargene 20000" was used to define the terms "upstream" and "downstream" as regions located 20 kilobases away from the transcription start site or transcription end site. To test whether the lead SNPs from a specific functional category is enriched, we compared lead SNPs against all SNPs within the whole genome. Fold enrichment(*E*) is calculated from the proportion of SNPs with a certain annotation divided by the proportion of SNPs with the same annotation in the background. We performed two-sided Fisher's exact test for each category of annotations to test if the fold enrichment is higher than expected.

To enable comparison of effect size across different studies, regardless of the direction of effect, we converted *P*-values into Z-statistics (two-sided) and expressed the standardized effect size (*β*) as a function of MAF and sample size, as described previously[73], using the following equation:

$$\beta = \frac{z}{\sqrt{2p(1-p)(n+z^2)}}, \text{s.e.m} = \frac{1}{\sqrt{2p(1-p)(n+z^2)}}, \quad (1)$$

where *p* is MAF, and *n* is the total sample size. We used the MAF of an ancestry-matched reference panel. s.e.m is the standard error of the mean.

## Fine mapping of GWAS loci

The fine-mapping analysis was performed on our curated cancer GWAS summary statistics with ancestry-matched-LD information using a recent toolkit[74] integrating three fine-mapping methods of PAINTOR (v.3.0)[75], CAVIARBF (v.0.2.1)[76], and FINEMAP (v.1.3.1)[77]. We only allow each causal block contains only one causal variant, and we used the recommended parameters for these tools. These fine-mapping tools provide the posterior inclusion probability (PIP) of each variant being the causal one within a specific model. Subsequently, we identified credible sets comprising variants with cumulative PIP values exceeding a threshold 0.95.

## SNP heritability estimation and genetic correlation

We estimated the SNP heritability($h_g^2$), which represents the proportion of phenotypic variance that can be attributed to common SNPs of each cancer based on both genotyped and imputed SNPs using LDSC (v1.0.1)[34]. Briefly, we first convert cancer GWAS summary statistics to the.sumstats format using "munge_sumstat.py". LDSC used HapMap3 SNPs with the option "--merge-alleles w_hm3.snplist", where w_hm3.snplist is the list of SNPs and alleles to ensure the alleles in our summary statistics files match those in the data used to estimate LD

scores. We excluded the SNPs of disproportionately large effect (i.e $c^2 > 80$) compared to the rest of the genome with the flag "--chisq-max 80" as recommended[78]. We then calculate the heritability using the formatted cancer.sumstats files with pre-calculated ancestry-matched reference LD scores, which were obtained from https://alkesgroup.broadinstitute.org/LDSCORE/. Furthermore, genetic correlation($r_g$) between pairwise cancers was estimated with the 1000 Genomes Project reference panel using LDSC (v1.0.1). The $r_g$ estimated by LDSC is an unbiased estimate and may exceed [-1,1] when standard errors are large, and the genetic correlation between studies is high. Genetic correlations for which the *P*-value survived the correction for multiple testing, with Bonferroni-corrected *P* < 0.05, were considered significant.

## Enrichment of molecular QTLs within GWAS risk loci

We used *fgwas* (v.0.3.6)[36] to assess the enrichment of molecular QTLs within GWAS risk loci. Briefly, GWAS loci were annotated as 3'aQTLs (or sQTLs, eQTLs) in a binary manner. We considered all molecular QTLs that were significant (FDR < 5%). *fgwas* then constructed a hierarchical Bayesian model to estimate the enrichment effects of different molecular annotations within GWAS loci. Additionally, the quantile–quantile plots (Q-Q plots) were used to visualize the *P*-values of cancer GWAS SNPs.

We further applied stratified LD score regression (v1.0.1)[37,38] to cancer GWAS results to assess the enrichment of heritability attributable to 3'aQTLs, sQTLs, and eQTLs within GWAS risk loci. Briefly, we included the functional categories in the "baseline-LD model" with 53 other functional categories[37]. We then created binary annotations for 3'aQTLs, sQTLs, and eQTLs, respectively, that is, we assigned an annotation value of 1 to the most significant 3'aQTL and a value of 0 to the remaining SNPs. We computed the LD scores of the SNPs using genotype data from individuals of European ancestry from the 1000 Genome Project (phase 3) with a window size of 1 cM. The heritability enrichment of a category was calculated as the proportion of heritability explained by the category divided by the proportion of SNPs in the category.

## Quantification of APA levels using DaPars2

We utilized our previously developed DaPars2 software[23,26], to calculate the poly(A) site-usage index (PDUI) value through a joint analysis of multiple samples employing a two-normal mixture model. The procedure involved several steps: Firstly, we extracted a 3'UTR annotation for each gene using the "DaPars_Extract_Anno.py" script within DaPars2. Subsequently, the sequencing depth for each sample was calculated using the "samtools flagstat" command. Finally, the PDUI value of each transcript across the samples was computed using DaPars2.

## 3'aQTL mapping and fine-mapping

We performed comprehensive 3'aQTL mapping across 49 human tissues from GTEx project and 18 tumor tissues from TCGA. Within each tissue, we transformed the PDUI values for each 3'UTR APA into quantiles of the standard normal distribution. To effectively address potential hidden batch effect and other unobserved covariates, our association analyses incorporated covariates. For GTEx data, these encompassed factors like the WGS sequencing platform, WGS library construction protocol and donor sex. Similarly, TCGA data analyses included covariates containing age, gender, and AJCC stage. Furthermore, we integrated the top 5 genotype principal components and PEER factors as essential components of the covariate set. The selection of PEER[79] factors was guided by established criteria from the GTEx consortium[24]: 15 PEER factors for tissue sample sizes <150, 30 PEER factors for sample sizes ranging from 150 to 250, and 35 PEER factors for sample sizes exceeding 250.

In each tissue, 3'aQTLs were identified via linear regression, employing Matrix eQTL[27], while accounting for aforementioned

covariates. Our analysis specifically focused on variants within a 1 Mb of the 3′UTR region, and minor allele frequencies ≥0.01 within the analyzed tissue. For the compilation of significant variant-APA pairs, we considered pairs with a false-discovery rate (FDR) below 0.05 as significant, and these were designated as 3′aQTL.

We further used a fine-mapping approach to identify candidate causal variants that underlie *cis*-aQTL loci. We identified 95% credible set variants using CAVIAR[76] (v.2.2) software and allowing for two causal variant with flag "-*c* 2". LD information between SNP pairs was generated using PLINK (v1.90)[70].

## Colocalization tests

To determine whether the association of cancer GWAS SNP is mediated through the regulation of molecular QTL, we performed colocalization analysis using the *coloc* R package (v.4.0.4)[80] with default priors. For each cancer GWAS trait, we first identified the lead SNP, simply characterized by a SNP with *P*-value $< 5 \times 10^{-8}$, and positioned more than 1 Mb away from other variants exhibiting higher statistical significance. Subsequently, for each lead SNP, we compiled the list of all features (gene/intron cluster/transcript) located within 1 Mb radius for colocalization analysis. Utilizing the *coloc* method, we calculated five distinct posterior probabilities (PPs): PP0, signifying a null model of no association; PP1, representing exclusive genetic association at the GWAS SNP; PP2, denoting sole genetic association at the 3′aQTL; PP3, indicating concurrent association at both the GWAS SNP and 3′aQTL, albeit with differing causal variants; PP4, reflecting association at both the GWAS SNP and 3′aQTL, underpinned by a shared causal variant. To pinpoint instances of colocalization, transcripts were designated as colocalization events if PP4 ≥ 0.75 and PP4/(PP4 + PP3) ≥ 0.9, following the our prescribed criteria[23]. These identified colocalization events were then visually represented. LocusZoom (v.1.4)[81] was used for region visualization plots, and PLINK (v.1.90)[70] was used for calculating LDs between identified causal SNP and other SNPs. For each colocalized gene, we defined the gene biotypes from human reference genome from NCBI (https://www.ncbi.nlm.nih.gov/assembly/GCF_000001405.39/) and GENCODE v26 GTF. The "protein-coding genes" group included any genes with the "protein_coding" biotype in the GTF file. While the "lncRNA genes" group included any genes with a long noncoding gene biotype ("lncRNA", "processed_transcript", "ense_intronic", "sense_overlapping", "antisense", "macro_lncRNA", "bidirectional_promoter_lncRNA", "3prime_overlapping_ncRNA") in GTF file.

## APA TWAS for cancer GWAS

We employed FUSION[42] to conduct TWAS for APA, with a primary focus on individuals of European ancestry. Our approach commenced with the implementation of a mixed-linear model to estimate the heritability of the 3′UTR region. This estimation employed SNPs with MAF > 0.01, situated within a 1 Mb of the 3′UTR of each gene, within a reference panel comprising cohorts featuring matched RNA-seq and genotype data. To ensure robust covariate adjustment, well-established factors used in QTL mapping section were incorporated to determine residualized PDUI values. Subsequently, only transcripts exhibiting significant heritability estimates ($cis$-$h^2$) below a Bonferroni-corrected *P*-value threshold of 0.05 were retained for subsequent analysis (Fig. S13a). From the array of methodologies available within the FUSION framework, four different models were chosen for weight calculation: best linear unbiased predictor (BLUP), elastic-net regression (Elastic Net), lasso regression (LASSO), and single best eQTL (Top1). A cross-validation approach was employed to select the model demonstrating the optimal 3a′TWAS prediction accuracy for each gene. We then applied 3′aTWAS prediction models to GWAS summary statistics, employing an FDR threshold of 0.05. For comparative analysis, we also conducted TWAS for gene expression and splicing using

the same GWAS summary statistics. The expression and splicing TWAS models of GTEx were obtained from PredictDB[82].

## Joint conditional probability analysis

We performed joint and conditional analyses using FUSION, which used an iterative process that progressively includes predictors into the model until no significant associations were observed. These analyses were carried out using the default locus window size of 100,000 base pairs. Briefly, we conducted testing for 3′aTWAS significant associations (FDR < 0.05) to evaluate the independence of associations within their respective 1-Mb window. The analysis was performed on all candidate hit regions using the "FUSION.post_process.R" script, allowing us to disentangle signals within regions containing multiple significant genes and generate conditional output plots. This analysis assesses the probability of multiple associations occurring simultaneously (jointly) and helps distinguish between genes associated independently (marginal) from those dependent on surrounding loci (conditional).

## Annotation of 3′aTWAS-identified genes in cancer-relevant gene databases

To identify the overlap between our 3′aTWAS-identified genes and known cancer-related genes, we collected cancer-related gene sets from the Molecular Signatures Database (MSigDB)[57], DORGE[55], and CGC genes[56] from the COSMIC website (https://cancer.sanger.ac.uk/census). Putative cancer-related genes were identified by annotating with specific key phrases, such as "breast cancer" and "prostate cancer".

## Effects of CRISPR−Cas9 gene silencing on proliferation in cancer-relevant cells

Gene-dependency levels for 17,386 genes based on CRISPR-Cas9 essentiality screen datasets, including "CRISPR_gene_effect.csv" as determined by the CERES computational method and "sample_info.csv", were downloaded from the Dependency Map (DepMap) portal(https://depmap.org/portal/download/all/) public 22Q2[47]. A cutoff CERES value < −0.5 was used to determine essentiality. For each gene, we calculated the significance on cell proliferation based on the count of CERES values < −0.5 in the total count of relevant cancer cells, using the Binomial test. Additionally, the project deep RNAi interrogation of viability effects in cancer (DRIVE)[49] was leveraged to ascertain the essentiality of genes. For each gene, we obtained the redundant siRNA activity (RSA) score and applied a threshold of -3 to determine its classification as essential or non-essential.

## Dual-luciferase reporter construction

Renilla luciferase in psiCHECK-2 Vector (Promega, cat#: C8021) was used as a primary reporter gene. The firefly reporter cassette served as an intra-plasmid transfection normalization reporter[83]. To investigate the role of 3′aQTLs in regulating their target genes, the 3′UTR containing reference and alternative variants of *CRLS1* was cloned downstream of the Renilla luciferase translational stop codon of psiCHECK-2 Vector, respectively. To investigate the role of poly(A) site usage in regulating target genes, the short 3′UTR and long 3′UTR with mutated proximal PAS sequence of *CRLS1, AMFR, ATG10, RPAIN* was cloned downstream of the Renilla luciferase translational stop codon of psiCHECK-2 Vector, respectively.

## Rapid amplification of cDNA ends (RACE)

MCF-7 cell lines obtained from the Cell Resource Center of Shanghai Institutes for Biological Sciences (Chinese Academy Science, Shanghai, China) were cultured in Dulbecco's Modified Eagle Medium (DMEM; Gibco, cat #: C11995500BT), containing 10% fetal bovine serum (FBS; Gibco, cat #: C10010500BT), 100-IU/mL penicillin, and 100-µg/mL

streptomycin (Gibco, cat #: 15140-122). Cells were cultured at 37 °C in a 5% $CO_2$ atmosphere with 100% humidity. To begin with, the previously described luciferase psiCHECK-2 vector (constructed by Tsingke biotech and confirmed by sequencing) were transfected into cells using Lipofectamine 3000 Transfection Reagent (Invitrogen, cat#: L3000015). After 48 h, cells were harvested for RNA extraction. Total RNA was extracted using TRIzol reagent (Sigma, cat#: T9424) according to the manufacturer's instruction. Next, the full length of 3′ UTR was identified and amplified from the total RNA using the GoScript™ Reverse Transcriptase (Promega, cat#: A5001) with oligo dT18-XbaKpnBam primer following the manufacturer's protocol. Finally, 3′ RACE was carried out using the check-luc forward primer and XbaKpnBam reverse primer to distinguish exogenous RNA from endogenous RNA. RACE-PCR products were separated on a 1% agarose gels. Bands were excised and the extracted using TIANgel mini purification kit (Tiangen, cat#: DP209) were cloned into PCE2 vector by 5 min TOPO-Blunt Cloning Kit (Vazyme, cat#: C602) for Sanger sequencing[84]. The primers used for 3′ RACE are listed in Supplementary Data 9.

## Dual-luciferase reporter assay
MCF-7 cells were seeded in 1 day prior to transfection. The previously described luciferase psiCHECK-2 vector were transfected into cells. Forty-eight hours post-transfection, firefly and renilla luciferase activities were measured by Dual-Luciferase Assay System (Promega, #E1980) on a BioTek Synergy H1 plate reader with full waveband. Each assay was measured in three independent replicates.

## Gene knockdown with siRNA and quantitative RT-PCR
When MCF-7 cells reached 60–80% confluency, they were transfected with NTC siRNA or a pool of three different gene-specific siRNAs (Supplementary 9; RiboBio), using Lipofectamine RNAiMAX Reagent (Invitrogen, cat #: 13778150), at a final concentration of 50 nM, according to the manufacturer's protocol. The medium was replaced after 12 h, and cells were harvested 48 h after transfection.

Total RNA was extracted using the Quick-RNA™ Miniprep Kit (cat#: R1055; Zymo Research), and cDNA was generated using the Hifair® III 1st Strand cDNA Synthesis Super Mix for qPCR (gDNA digester plus) kit (Yeasen, cat #: 11141ES60). Quantitative PCR was performed using the Hieff® qPCR SYBR Green Master Mix (Yeasen, cat #: 11203ES08) on a CFX96 machine (BIO-RAD, Hercules, CA, USA). Primers used for qPCR are listed in Supplementary Data 9. Experiments measuring expression of each gene were repeated at least three times, with *GAPDH* used as the internal reference for expression.

## Cell proliferation assay
Cell proliferation assays were performed using the CellTiter-Glo 2.0 Kit (Promega, cat #: 92243). Briefly, MCF-7 cells were transfected with siRNA for 16–20 h. A total of 5000 cells from each treatment were then re-plated into 96-well plates in quadruplicates. After 72 h, the medium was replaced with 200 μL of a 1:1 mixture of DMEM and CellTiter-Glo 2.0 reagent. The cells were lysed on an orbital shaker at 300 rpm for 2 min at room temperature. The plates were equilibrated for 10 min, and the luminescent signal, reported as relative light units (RLU), proportional to the amount of ATP, was measured at the full waveband with a BioTek Synergy H1 plate reader. Results were calculated by GraphPad Prism v.8.0.1 (GraphPad, San Diego, CA, USA).

## Cell colony formation assay
MCF-7 cells were transfected with siRNA for 16–20 h, and 2000 cells were then re-plated into 12-well plates in triplicates for each gene. When single clones contained more than 50 cells, the colonies were fixed with 4% paraformaldehyde for 20 min and stained with 0.1% crystal violet (Sangon Biotech, cat #: A600331-0025) for 20 min at room temperature. After washing with water, the plate was air dried

before imaging. Colony counting was performed using ImageJ 1.53r (National Institutes of Health, Bethesda, MD, USA).

## Generation of stable knockdown cell line using lentivirus delivered shRNA
Two set of shRNAs against each gene was used to cloned into pLKO.1-puro vector (Supplementary Data 9). shRNA-expressing lentivirus was produced with the third-generation packaging system in human embryonic kidney (HEK) 293T cells (Cell Resource Center of Shanghai Institutes for Biological Sciences). Briefly, 70–80% confluent 293T cells in 6-well plate were transiently cotransfected with 5 μg of lentiviral transfer vector, 1.67 μg of pVSVG (envelope plasmid), 1.67 μg of pRSV-Rev (packaging plasmid) and 1.67 μg of pMDLg/pRRE (packaging plasmid) with PEI according to the manufacturer's instructions. Medium was replaced 24 h after transfection with DMEM containing 10% FBS and 0.1% penicillin and streptomycin, and virus supernatant was collected every 24 h for up to 2 d. Supernatant containing viral particles was filtered through a 0.45-μm filter unit and stored at −70 °C in aliquots or used directly for cell infection. For lentivirus infection, target cells were seeded in a 6-well plate or a 10-cm dish 16–18 h before infection and were grown to 70–80% confluency upon transduction. Culture medium was removed, and cells were incubated with virus supernatant along with 8 μg/ml polybrene. After overnight incubation, the virus-containing medium was replaced with a fresh medium. Puromycin was applied to kill non-infected cells 36 to 48 h after infection. After two days of selection, when non-infected control cells were all dead, surviving cells were split and maintained with the same concentration of puromycin. After 3 d, cells were collected for RNA and tested by RT-qPCR to confirm the successful shRNA knockdown efficiency of target genes.

## Cell viability and proliferation assays for shRNA-mediated knockdown
Cells were trypsinized, resuspended at $1 \times 10^4$ cells/ml, and seeded in 96-well plates, with each well containing 100 μl medium of $1 \times 10^3$ cells. Cell viability and proliferation were determined using CCK8 assays (Yeasen, cat#:40203ES76) at designated time points by measuring the absorbance at 450 nm, following the manufacturer's instructions. Values were obtained from four replicate wells for each treatment and time point. Results are representative of three independent experiments.

## Analysis of PPI network and pathway enrichment
We selected the pan-cancer identified 3′aTWAS genes and CGC genes, filtering for those that encode for HLA genes within MHC regions. Functional pathway enrichment analysis for non-HLA 3′aTWAS genes was performed using the Database for Annotation, Visualization, and Integrated Discovery (DAVID, v.2021)[85]. Furthermore, we calculated the interconnectivity of these non-HLA 3′aTWAS genes with CGC genes based on physical PPIs using STRING (v.12.0)[86] with default parameters, confidence score cutoff is 0.4, and maximum additional interactors of 5. To visualize the PPI network, enriched diseases, and pathways associated with 3′aTWAS genes, we utilized Cytoscape (v.3.9.1)[87].

## A comprehensive data portal to host cancer susceptibility genes
We constructed the Cancer GWAS Database (TCGD) (http://bioinfo.szbl.ac.cn/TCGD/index.php) as a comprehensive data portal for cancer susceptibility gene exploration, allowing users to browse, search, and visualize crucial information. TCGD has included 55 cancer GWAS summary statistics across 22 cancer types, which are provided for researchers and users to investigate the role of APA in human cancer. The users can explore cancer GWAS summary statistics in the "Browse GWAS" module by selecting cancer/neoplasm type, population type, published year, and total sample size. Meanwhile, all the columns in

this summary table can be searchable and sorted based on the demand of customers. In addition, they can list each GWAS with the Q-Q plot, Manhattan plot, risk loci and SNP heritability, and other detailed information. The multi-tissue TWAS data is also organized in a tabular format. The users can browse and search their genes (e.g., *IRF5*) or information (e.g., 3'aQTL) for the multi-tissue TWAS. Then, clicking on a gene name provides access to browsing and searching the single-tissue TWAS.

## Reporting summary

Further information on research design is available in the Nature Portfolio Reporting Summary linked to this article.

## Data availability

Raw whole transcriptome and genome sequencing data from the Genotype-Tissue Expression (GTEx) project are available via the database of Genotypes and Phenotypes (dbGaP) under the accession number: phs000424.v8.p2[24]. All processed GTEx data are available via the GTEx portal (http://gtexportal.org/). Publicly RNA-seq and genotype data from The Cancer Genome Atlas (TCGA) from the Genomic Data Commons (GDC) Data Portal (https://portal.gdc.cancer.gov/). Expression data profiles were obtained from the Xena 2 (https://tcga.xenahubs.net). PDUI data profiles were obtained from the TC3A (http://tc3a.org). GWAS summary statistics are from NHGRI–EBI GWAS catalog (https://www.ebi.ac.uk/gwas/), UK Biobank GWAS (http://www.nealelab.is/uk-biobank/), Finn Gen (https://www.finngen.fi/en) and JENGER(http://jenger.riken.jp). The details, including accession numbers, of GWAS summary statistics used in this study, are listed in Supplementary Data 1. 1000 Genomes Project Reference for LDSC, https://data.broadinstitute.org/alkesgroup/LDSCORE/1000G_Phase3_plinkfiles.tgz; 1000 Genomes Project Reference with regression weights for LDSC, https://data.broadinstitute.org/alkesgroup/LDSCORE/1000G_Phase3_weights_hm3_no_MHC.tgz. All significant 3' aTWAS genes in cancer are available in Supplementary Data 5. The expression and splicing TWAS models for GTEx v8 are publicly available at PredictDB (https://predictdb.org/). Source data are provided with this paper.

## Code availability

The custom source codes to perform the data analysis relevant to this paper are available, under the MIT license, on Zenodo with the access code https://doi.org/10.5281/zenodo.8223680 and Github https://github.com/lilab-bioinfo/CancerAPA.

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

## Acknowledgements

We thank members of the Li laboratory for helpful discussions. This work was supported by the National Key Research and Development Program of China (no. 2022YFA1302800) to L.D., National Natural Science Foundation of China (no. 32100533, 32370721) and Open grant funds from Shenzhen Bay Laboratory (no. SZBL2021080601001), to L.L. National Natural Science Foundation of China (no. 32270779) to L.D. We also thank Qin Wang at Shenzhen Bay Laboratory supercomputing center for high-computing support. Work of M.P. is supported by a Ramón y Cajal contract (RYC2018-024564-I). MP thanks CERCA Program/Generalitat de Catalunya for IDIBELL institutional support.

## Author contributions

L.L., D.L., and W.L. conceived and supervised the project. H.C., Q.W., W.C., R.D., L.G., X.L., X.Z. performed the bioinformatics analysis. Z.W., and J.W. C.L., performed the experiments. X.M. constructed the website. M.P., T.N., G.-H.,. W contributed expertise in cancer and genetic analyses. H.C., Z.W., and L.L. wrote the manuscript with assistance from other authors.

## Competing interests

The authors declare no competing interests.

## Additional information

¹Institute of Systems and Physical Biology, Shenzhen Bay Laboratory, Shenzhen 518055, China. ²Institute of Molecular Physiology, Shenzhen Bay Laboratory, Shenzhen 518055, China. ³Gene Regulation of Cell Identity Group, Regenerative Medicine Program, Bellvitge Institute for Biomedical Research (IDIBELL), L'Hospitalet de Llobregat, Barcelona 08908, Spain. ⁴Program for Advancing Clinical Translation of Regenerative Medicine of Catalonia, P-CMR[C], L'Hospitalet de Llobregat, Barcelona 08908, Spain. ⁵Center for Networked Biomedical Research on Bioengineering, Biomaterials and Nanomedicine (CIBER-BBN), Madrid 28029, Spain. ⁶Department of Biochemistry and Molecular Biology of School of Basic Medical Sciences, Shanghai Medical College of Fudan University, Shanghai 200032, China. ⁷State Key Laboratory of Genetic Engineering, Collaborative Innovation Center of Genetics and Development, Human Phenome Institute, School of Life Sciences and Huashan Hospital, Fudan University, Shanghai 200438, China. ⁸Disease Networks Research Unit, Faculty of Biochemistry and Molecular Medicine & Biocenter Oulu, University of Oulu, Oulu 90410, Finland. ⁹Division of Computational Biomedicine, Department of Biological Chemistry, School of Medicine, The University of California, Irvine, CA 92697, USA. ¹⁰These authors contributed equally: Hui Chen, Zeyang Wang. ✉e-mail: wei.li@uci.edu; denglin@szbl.ac.cn; lei.li@szbl.ac.cn

