## [Peer Review File · Nature Communications]

A distinct class of pan-cancer susceptibility genes revealed by an alternative polyadenylation transcriptome-wide association studyEditorial Note: Parts of this Peer Review File have been redacted as indicated to remove third-party material where no permission to publish could be obtained.

REVIEWER COMMENTS

Reviewer #1 (Remarks to the Author):

In this study, the authors conducted a comprehensive pan-cancer analysis by integrating cancer GWAS data from 22 cancer types and genetic variants associated with APA (3'aQTL) across human tissues sourced from GTEx and TCGA. They integrated a large collection of cancer GWAS summary statistics from various sources, providing valuable data for the cancer research community. Subsequently, they employed colocalization analysis and transcriptome-wide association study to pinpoint APA-related cancer susceptibility genes. To validate their findings, they demonstrated the APA of CRLS1 gene increased protein abundance and promote breast cancer cell proliferation. Additionally, they developed a publicly accessible database for the research community, greatly enhancing the utilization of these valuable data. This study discovers APA-related cancer susceptibility genes and enhances our understanding of human cancer etiology.

Here are some recommendations and suggestions to address:

1. Line 194: For the sentence "our findings indicate that 3'aGenes are largely distinct from eGenes and sGenes across many cancer types." Please provide supplementary data or figures to support this finding.
2. Line 218-224: please ensure the appropriate citation of related papers for these examples. Also thoroughly review the rest of the manuscript to avoid any potential citation omissions.
3. Lines 258-260: it would be valuable to supplement the results to showcase the list of genes that identified both in colocalization and 3'aTWAS analyses, as well as those identified in both GTEx and TCGA datasets.
4. As the development of the TCGD database represents a significant contribution to the research community, it is recommended to include a dedicated result section that introduces the database's functions and provides guidance on how users can effectively utilize each module.
5. In the GWAS module on the website, it currently only displays a list of GWAS studies without providing any GWAS summary data for users to browse and download.
6. It is necessary to implement a search bar on the website to facilitate user access to results based on input or selection of gene names, tissues, mQTL types, and other parameters.
7. The website's download page currently offers data only for "3'aTWAS models of 49 tissues in GTEx v8 cohort." It's necessary to include data for other modules, such as cancer GWAS summary data, colocalization results and TWAS results in TCGA samples.
8. Please check if the 'help' page on the website is functioning correctly, as it seems to not be working.
9. The use of SMR in colocalization is mentioned on the website but not mentioned in the paper. Please address this to ensure consistency.
10. Please round decimal numbers on the website to enhance readability and user-friendliness.

Reviewer #2 (Remarks to the Author):

In this manuscript, Chen et.al have made a significant contribution by identifying a distinct class of pan-cancer susceptibility genes through an comprehensive analysis of alternative polyadenylation transcriptomes. To achieve this, the authors meticulously collected and integrated GWAS summary statistics from three reputable sources, including the UK Biobank and BioBank Japan. This effort resulted in a valuable dataset encompassing 80 well-powered GWAS datasets across 23 major cancer types, alongside APA quantification from a staggering 23,955 RNA sequencing samples involving 7574 individuals.

The crux of their investigation involved conducting a transcriptome-wide association study (TWAS) to pinpoint genes linked to cancer susceptibility. Impressively, their study unveiled a robust set of 1,178

genes that exhibited significant associations with cancer susceptibility across multiple tumor types. Notably, these genes displayed enrichments in functions pertaining to DNA repair, cell cycle regulation, and immune response. Furthermore, the authors have provided a user-friendly hub, offering valuable resources to the research community for decoding the functional aspects of cancer risk variants and exploring the genetic underpinnings of a variety of human cancers. Significantly, the manuscript reports experimental validation through an in vitro experiment, confirming the role of an allele in breast cancer susceptibility risk.

The findings of this study hold potential to impact the cancer research and treatment by providing new targets for drug development and personalized cancer therapies. The manuscript is commendably well-structured and well-written, with an abundance of informative results. Overall, this manuscript represents an impressive work with substantial implications for cancer research and treatment. The methods are sound and well documented.

I have only a few minor points to be addressed to improve the manuscript.

The font size for gene names in Fig.1f-g, Fig.4a, Fig.S10b, S11, S13, and others appears to be too small and should be enhanced for clarity.

Some figure citations lack specificity. For instance, in lines 115-119, Fig.S4 comprises four panels, but the manuscript only references "Fig.S4." It would be beneficial to specify which panel(s) are being referred to in the text.

In Fig.4f-g, it is observed that the G to A allele is associated with APA events, and longer UTRs are linked to higher protein levels. However, given that longer UTRs typically correspond to lower RNA stability, it would be valuable for the authors to provide commentary or discuss the reasons behind the observed increase in protein levels.

Regarding the TCGD database, I encountered a permission issue when attempting to download the dataset, said "You don't have permission to access this resource". I recommend addressing this issue and ensuring that the necessary permissions are granted before the manuscript is accepted.

Reviewer #3 (Remarks to the Author):

In this study, the authors preformed the first large-scale and systematic analysis assessing APA-mediated genetic effects across the cancer spectrum. They identify 642 cancer susceptibility genes predicted to modulate cancer risk via APA, and functionally evaluate one such variant (CRLS1) in mediating breast cancer risk. Overall, the data presented are robust and the conclusions are generally supported by the evidence provided. This work is particularly significant for the resource it provides the community, specifically, a large catalog of common genetic variants potentially regulating APA that may promote cancer imitation. Below are some specific comments that I feel would strengthen the manuscript:

- 1) The authors find that breast and prostate cancers have the greatest number of APA-linked susceptibility genes (Fig 4b). Is this due to more patient samples from these tumor types, or an intrinsic biological difference?
- 2) In Fig 4, the authors claim that APA-linked susceptibility genes tend to have greater impact on cancer cell proliferation, relative to non-3'aTWAS genes. However, is this the case when compared to splicing and expression TWAS genes?
- 3) It is unclear how the 3'RACE experiment was performed. Was this detecting exogenous DNA? Please draw arrows next to the gel highlighting the expected short and long forms (Fig 5e).

4) While the authors clearly demonstrate that the long form of CRLS1 3'UTR drives increased protein expression, and that CRLS1 mediates breast cancer cell growth (to a modest extent), there is no evidence that the specific variant contributes to cell proliferation (line 344). To support this conclusion, the authors could overexpress both the WT and variant forms in non-transformed mammary epithelial cells and measure cell proliferation. Or the authors could alter their conclusions.

5) On lines 333-335, the authors make a bold claim that these variants play a significant role in cancer initiation. While this may be true, this is not tested experimentally in their manuscript. I would suggest toning down this statement.

Reviewer #1 (Remarks to the Author):

In this study, the authors conducted a comprehensive pan-cancer analysis by integrating cancer GWAS data from 22 cancer types and genetic variants associated with APA (3'aQTL) across human tissues sourced from GTEx and TCGA. They integrated a large collection of cancer GWAS summary statistics from various sources, providing valuable data for the cancer research community. Subsequently, they employed colocalization analysis and transcriptome-wide association study to pinpoint APA-related cancer susceptibility genes. To validate their findings, they demonstrated the APA of CRLS1 gene increased protein abundance and promote breast cancer cell proliferation. Additionally, they developed a publicly accessible database for the research community, greatly enhancing the utilization of these valuable data. This study discovers APA-related cancer susceptibility genes and enhances our understanding of human cancer etiology.

Response: We thank the Reviewer for acknowledging the significance and novelty of our findings and also for an excellent summary of our revised manuscript.

Here are some recommendations and suggestions to address:

1. Line 194: For the sentence "our findings indicate that 3'aGenes are largely distinct from eGenes and sGenes across many cancer types." Please provide supplementary data or figures to support this finding.

Response: We thank the reviewer for the valuable suggestions. Our initial version stated that "our findings indicate that 3'aGenes are largely distinct from eGenes and sGenes across many cancer types". This was based on the observation that the 3'aGenes we identified were enriched in the different pathways compared to the eGenes and sGenes (Figure 3f-g). Unlike that, the eGenes and sGenes were enriched in the pathway related to DNA repair and chromatin organization processes. 3'aGenes were found to be involved in protein localization. To further support this finding, we have now conducted more analysis of these genes in the revised manuscript and found the sequence structure characteristics of colocalized 3'aGenes exhibited differences when compared to eGenes and sGenes. While 3'aGenes shared comparable 5'UTRs length with eGenes and sGenes, they exhibited relatively shorter coding region (CDS) when compared with eGenes ($P = 1.75 \times 10^{-2}$) and sGenes ($P = 1.87 \times 10^{-3}$, New Fig. S11), and much longer 3'UTR lengths compared with eGenes ($P = 2.47 \times 10^{-15}$) and sGenes ($P = 8.82 \times 10^{-25}$, New Fig. S11). Furthermore, we found a significantly higher prevalence of adenylate-uridylate-rich (AU-rich) elements proximal to poly(A) sites in colocalized 3'aGenes in comparison to eGenes ($P = 6.43 \times 10^{-9}$) and sGenes ($P = 3.04 \times 10^{-9}$), suggesting that 3'aGenes harbor an increased number of potentially regulatory elements. (New Fig. S12).

New Figure S11. Sequence structure comparisons between colocalized 3'aGenes, eGenes and sGenes. *P*-values were determined through a two-sided t-test. The center horizontal lines of the box plot show the median values, and the boxes span from the 25th to the 75th percentile. n.s., not significant.

New Figure S12. Colocalized 3'aGenes have more AU-rich elements proximal to poly(A) sites than eGenes and sGenes. a. Boxplot showing the number of overall AU-rich elements proximal to poly(A) sites (ranged from -100 to 100 bp from the poly(A) site) in each gene, comparing 3'aGenes with eGenes and sGenes. The center horizontal lines of the Boxplot represent median values, and the boxes span from the 25th percentile to the 75th percentile. Whiskers extend to $1.5 \times$ IQR (bottom), where IQR is the interquartile range. **b.** Boxplots for each specific motif variant proximal to poly(A) sites in 3'aGenes, eGenes, and sGenes.

2. Line218-224: please ensure the appropriate citation of related papers for these examples. Also thoroughly review the rest of the manuscript to avoid any potential citation omissions.

Response: We thank the reviewer for the suggestion. We have now included the appropriate citation of related papers for the examples in the revised manuscript. We have also further made a thorough review of the manuscript to avoid potential citation omissions.

“Interestingly, our 3'aTWAS identified multiple known and novel cancer risk genes, such as small G protein signaling modulator 3 (SGSM3)¹, which is significantly associated with breast cancer in breast mammary tissue ($P_{3'aTWAS} = 2.6 \times 10^{-18}$, $P_{eTWAS}=0.47$, $P_{sTWAS}=0.40$). This finding suggests that 3'UTR APA usage of SGSM3, rather than the expression or the splicing of the SGSM3 gene, mediates breast cancer risk.”

“... eight genes exhibited similar or even higher levels of essentiality compared to the well-known oncogene MYC²...”

“... a known APA regulator that potentially modulates the biological effect of rs2235816 (Fig. S20)³.”

“RACE-PCR products were separated on a 1% agarose gels. Bands were excised and the extracted using TIANGel mini purification kit (Tiangen, cat#: DP209) were cloned into PCE2 vector by 5 min TOPO-Blunt Cloning Kit (Vazyme, cat#: C602) for Sanger sequencing⁴.”

3. Lines 258-260: it would be valuable to supplement the results to showcase the list of genes that identified both in colocalization and 3'aTWAS analyses, as well as those identified in both GTEx and TCGA datasets.

Response: Thank you for your suggestions. We have added the list of genes identified in both GTEx and TCGA datasets in the updated **Supplementary Table S6-S7**. Briefly, the table now includes 47 genes that were identified both in the colocalization and 3'aTWAS analyses. Additionally, seven genes were consistently identified in 3'aTWAS analysis for both GTEx and TCGA datasets.

4. As the development of the TCGD database represents a significant contribution to the research community, it is recommended to include a dedicated result section that introduces the database's functions and provides guidance on how users can effectively utilize each module.

Response: We thank the reviewer for the valuable suggestion. We have now included a step-by-step tutorial in the “Help” module of the TCGD database, which provides users with an introduction to the database's functions and guidance on how to use them efficiently.

[redacted]

Review Figure 1. The screenshot displays the interface of the “Help” module. The user can find the “Help” at the end of the navigation bar and the step-by-step tutorial slides for the TCGD database.

5. In the GWAS module on the website, it currently only displays a list of GWAS studies without providing any GWAS summary data for users to browse and download.

Response: We thank the reviewer for the suggestion. We have now implemented a new function on the website which allows users to browse and download GWAS summary datasets. Furthermore, when the users click on a specific cancer study, the relevant GWAS information, including the Q-Q plot, Manhattan plot, and fine-mapping results for that particular GWAS study will be presented.

[redacted]

Review Figure 2. The GWAS block viewer in TCGD provides a comprehensive interface. **(a)** Within the “Browse GWAS” module, users can access detailed GWAS summary statistics through a super link which includes **(b)** trait information, where users can find the link for downloading corresponding GWAS summary data in the end of “GWAS source”; **(c)** QQ-plot and **(d)** the Manhattan plot for the GWAS summary statistics, both available for download in jpg and pdf formats via dedicated buttons;**(e)**

the fine-mapping results are also presented for the significant SNPs at genome-wide significance level ($P < 5 \times 10^{-8}$), showcasing the posterior probability calculated from three fine-mapping methods (PAINTOR, FINEMAP, and CAVIARBF).

6. It is necessary to implement a search bar on the website to facilitate user access to results based on input or selection of gene names, tissues, mQTL types, and other parameters.

Response: We thank the reviewer's suggestions. We have now implemented new features that allow users to select gene names, tissues, and xQTL types (we updated the term "mQTL" to "xQTL" in TCGD website to make it consistent) within each result module (**Review Figure. S3a**). We also added a prominent search bar at the top of the "Home" page and filter options that enable users to select their preferred tissues and genes (**Review Figure. S3b**). The Help page has been updated accordingly to provide users with clear and practical guidance on leveraging these functionalities.

[redacted]

Review Figure 3. The search engines added in the TCGD database. **a.** The search bar and filter options have been implemented in each result module, allowing users to select genes, tissues, and xQTL types. The result page for colocalization is used as an example. **b.** A prominent search bar at the top of the “Home” page enables users to select their preferred tissues and genes.

7. The website's download page currently offers data only for "3'aTWAS models of 49 tissues in GTEx v8 cohort." It's necessary to include data for other modules, such as cancer GWAS summary data, colocalization results and TWAS results in TCGA samples.

Response: We thank the reviewer for the valuable suggestion. As per the suggestion, we have now added more comprehensive datasets for downloading in the download page, including GWAS summary statistics, the colocalization results, and TWAS results from GTEx and TCGA samples (Review Figure 4). Additionally, we have made the colocalization and TWAS results available for download on the respective search page. Users can download the entire results or specific results for specific genes. We believe that this update will enhance the user experience and provide more comprehensive data for the community.

a Download

1. Download page provides the 3'aTwas models of GTEx v8 cohort and TCGA cohort

Tissue type	Study	Sample Size	File Name
Adipose - Subcutaneous	GTEx	493	Adipose_Subcutaneous.tar.gz
Adipose - Visceral (Omentum)	GTEx	403	Adipose_Visceral_Omentum.tar.gz
Adrenal Gland	GTEx	201	Adrenal_Gland.tar.gz
Artery - Aorta	GTEx	339	Artery_Aorta.tar.gz
Artery - Coronary	GTEx	181	Artery_Coronary.tar.gz
Artery - Tibial	GTEx	490	Artery_Tibial.tar.gz
Brain - Amygdala	GTEx	120	Brain_Amygdala.tar.gz
Brain - Anterior cingulate cortex (BA24)	GTEx	137	Brain_Anterior_cingulate_cortex_BA24.tar.gz
Brain - Caudate (basal ganglia)	GTEx	174	Brain_Caudate_basal_ganglia.tar.gz
Brain - Cerebellum	GTEx	159	Brain_Cerebellum.tar.gz
Brain - Cerebellar Hemisphere	GTEx	190	Brain_Cerebellar_Hemisphere.tar.gz
Brain - Cortex	GTEx	185	Brain_Cortex.tar.gz

b

2. Download page provides the colocalization results of GTEx v8 cohort and TCGA cohort

Cancer type	Study	Sample Size	File Name
Adipose - Subcutaneous	GTEx	493	Adipose_Subcutaneous_coloc.txt
Adipose - Visceral (Omentum)	GTEx	403	Adipose_Visceral_Omentum_coloc.txt
Adrenal Gland	GTEx	201	Adrenal_Gland_coloc.txt
Artery - Aorta	GTEx	339	Artery_Aorta_coloc.txt
Artery - Coronary	GTEx	181	Artery_Coronary_coloc.txt
Artery - Tibial	GTEx	490	Artery_Tibial_coloc.txt
Brain - Amygdala	GTEx	120	Brain_Amygdala_coloc.txt
Brain - Anterior cingulate cortex (BA24)	GTEx	137	Brain_Anterior_cingulate_cortex_BA24_coloc.txt
Brain - Caudate (basal ganglia)	GTEx	174	Brain_Caudate_basal_ganglia_coloc.txt
Brain - Cerebellum	GTEx	159	Brain_Cerebellum_coloc.txt
Brain - Cerebellar Hemisphere	GTEx	190	Brain_Cerebellar_Hemisphere_coloc.txt
Brain - Cortex	GTEx	185	Brain_Cortex_coloc.txt

c

3. Download 55 GWAS summary file across 22 cancer types

CGID	Cancer/Neoplasm type	Source	Sample Size	File Name
CGS01	Basal cell carcinoma	EUR	361141	CGS01.gwas_summary.txt
CGS02	Breast	EUR	69626	CGS02.gwas_summary.txt
CGS03	Breast	EUR	139274	CGS03.gwas_summary.txt
CGS04	Breast	EAS	95283	CGS04.gwas_summary.txt
CGS05	Breast	EUR	107722	CGS05.gwas_summary.txt
CGS06	Cervical	EUR	100969	CGS06.gwas_summary.txt
CGS07	Colorectal	EAS	202807	CGS07.gwas_summary.txt
CGS08	Breast	EUR	106676	CGS08.gwas_summary.txt
CGS09	Colorectal	EUR	177028	CGS09.gwas_summary.txt
CGS10	Female genital organs	EUR	121885	CGS10.gwas_summary.txt
CGS11	Eye	EUR	361141	CGS11.gwas_summary.txt
CGS12	Female genital organs	EUR	102986	CGS12.gwas_summary.txt

Review Figure 4. Main content of the download page for **(a)** 3'aTwas models of GTEx v8 cohort and TCGA cohort; **(b)** the colocalization results of GTEx v8 cohort and TCGA cohort **(c)** 55 GWAS summary file across 22 cancer types.

8. Please check if the 'help' page on the website is functioning correctly, as it seems to not be working.

Response: We are very sorry for the inconvenience. We have now fixed the 'help'

page issue.

9. The use of SMR in colocalization is mentioned on the website but not mentioned in the paper. Please address this to ensure consistency.

Response: We thank the reviewer and have now removed SMR on the website to ensure consistency.

10. Please round decimal numbers on the website to enhance readability and user-friendliness.

Response: We thank the reviewer and have now updated the website to display all numbers with round decimal numbers.

Reviewer #2 (Remarks to the Author):

In this manuscript, Chen et.al have made a significant contribution by identifying a distinct class of pan-cancer susceptibility genes through an comprehensive analysis of alternative polyadenylation transcriptomes. To achieve this, the authors meticulously collected and integrated GWAS summary statistics from three reputable sources, including the UK Biobank and BioBank Japan. This effort resulted in a valuable dataset encompassing 80 well-powered GWAS datasets across 23 major cancer types, alongside APA quantification from a staggering 23,955 RNA sequencing samples involving 7574 individuals.

The crux of their investigation involved conducting a transcriptome-wide association study (TWAS) to pinpoint genes linked to cancer susceptibility. Impressively, their study unveiled a robust set of 1,178 genes that exhibited significant associations with cancer susceptibility across multiple tumor types. Notably, these genes displayed enrichments in functions pertaining to DNA repair, cell cycle regulation, and immune response. Furthermore, the authors have provided a user-friendly hub, offering valuable resources to the research community for decoding the functional aspects of cancer risk variants and exploring the genetic underpinnings of a variety of human cancers. Significantly, the manuscript reports experimental validation through an in vitro experiment, confirming the role of an allele in breast cancer susceptibility risk.

The findings of this study hold potential to impact the cancer research and treatment by providing new targets for drug development and personalized cancer therapies. The manuscript is commendably well-structured and well-written, with an abundance of informative results. Overall, this manuscript represents an impressive work with substantial implications for cancer research and treatment. The methods are sound and well documented.

Response: We thank the Reviewer for acknowledging the significance and novelty of our findings and an excellent summary of our manuscript.

I have only a few minor points to be addressed to improve the manuscript.

The font size for gene names in Fig.1f-g, Fig.4a, Fig.S10b, S11, S13, and others appears to be too small and should be enhanced for clarity.

Response: We thank the reviewer for the valuable suggestions. We have increased the font size of the gene names in the figures to ensure readability, including Fig. 1f-g, Fig.4a, Fig. S10b, Fig. S11, Fig.S13, Fig. S14, Fig. S15 in the revised manuscript.

Some figure citations lack specificity. For instance, in lines 115-119, Fig.S4 comprises four panels, but the manuscript only references "Fig.S4." It would be beneficial to specify which panel(s) are being referred to in the text.

Response: We are sorry for the confusion. We have now included the specified panels of Fig. S4 in the text and have now examined the figure citations throughout the manuscript:

"Our observation revealed that these lead SNPs tend to exhibit large effect sizes ($P = 1.07 \times 10^{-39}$, Wilcoxon rank-sum test, Fig. 1c), while their minor allele frequency (MAF) is evenly distributed (Fig. S4a) in comparison to the entire genome."

"For example, the prostate cancer lead SNP rs4245739 was identified in the 3'UTR MDM4 (Fig. S4b), which encodes a regulator of p53, and breast cancer lead SNP rs1386230 was also located in the 3'UTR of FGF10 (Fig. S4b), which encodes the fibroblast growth factor 10. The lead SNPs within 3'UTR regions have effect sizes comparable to those in other genomic regions (P -value = 0.114, Wilcoxon rank-sum test, Fig. S4c-d)."

In Fig.4f-g, it is observed that the G to A allele is associated with APA events, and longer UTRs are linked to higher protein levels. However, given that longer UTRs typically correspond to lower RNA stability, it would be valuable for the authors to provide commentary or discuss the reasons behind the observed increase in protein levels.

Response: APA can have different effects on mRNA stability and translation efficiency in different genes. It has been frequently described that longer 3'UTR isoforms could generate higher protein levels. Below are several representative examples:

1. Autophagy-related protein 1 (Atg1) and Atg8a produce more protein through the expression of their long 3'UTR isoforms than their short isoforms. This lengthening of the 3'UTR is regulated by phosphorylate *Cpsf6*, a subunit of *CFI*.

Phosphorylation of *CPSF6* increases its activity by promoting its nuclear localization and RNA-binding capacity⁵.

2. The diabetic nephropathy-associated genes with APA-regulated 3'UTR lengthening had higher protein translation because they possessed more RBP-binding sites and were more likely to be regulated by certain translational enhancer RBPs. The reduction of the RNA-binding capacity of *CSTF2* supposedly recapitulates the lengthening of the hallmark transcripts, affecting the subcellular localization rather than the mRNA stability, ultimately culminating in increased protein abundance⁶.
3. The long 3' UTR isoform of Uncoupling protein 1 (UCP1) contributes only 5–10% of the total Ucp1 mRNA level, but its deletion reduced UCP1 expression by 50–60%. Cytoplasmic polyadenylation element-binding protein 2 (CPEB2) binds to the long 3' UTR isoform of the UCP1, which increases translation efficiency and protein abundance of UCP1⁷ (Review Figure 5).

In our case of *CRLS1*, the longer 3'UTR isoforms provided more frequent RBP binding sites (**New Figure S23a-b**), indicating it was more likely to be regulated by certain RBPs. We indeed observed the U-rich motif for *CPEB2*⁸ on *CRLS1* 3' UTR region, which has been demonstrated previously⁷ could increase the protein abundance.

[redacted]

Review Figure 5. Cytoplasmic polyadenylation element-binding protein 2 (CPEB2) binds to the long 3'UTR (LU) isoform of the uncoupling protein 1 gene (Ucp1), which increases translation efficiency and protein abundance of UCP1. (Figure was adapted from Mitschka, S., Nat Rev Mol Cell Biol,2022)

New Figure S22. The RBP binding sites for *CRLS1* 3' UTR region. **a.** Counts for RBP

binding sites identified at 3'UTR region of *CRLS1*. 3' UTR_L, lengthening 3' UTR region; 3' UTR_S, short 3' UTR region. **b.** The frequency of the number of potential binding sites for each RBP.

Regarding the TCGD database, I encountered a permission issue when attempting to download the dataset, said "You don't have permission to access this resource". I recommend addressing this issue and ensuring that the necessary permissions are granted before the manuscript is accepted.

Response: We apologize for the inconvenience and have now fixed this issue.

Reviewer #3 (Remarks to the Author):

In this study, the authors performed the first large-scale and systematic analysis assessing APA-mediated genetic effects across the cancer spectrum. They identify 642 cancer susceptibility genes predicted to modulate cancer risk via APA, and functionally evaluate one such variant (*CRLS1*) in mediating breast cancer risk. Overall, the data presented are robust and the conclusions are generally supported by the evidence provided. This work is particularly significant for the resource it provides the community, specifically, a large catalog of common genetic variants potentially regulating APA that may promote cancer initiation. Below are some specific comments that I feel would strengthen the manuscript:

Response: We thank the Reviewer for acknowledging the significance and novelty of our findings and also for a nice summary of our manuscript.

1) The authors find that breast and prostate cancers have the greatest number of APA-linked susceptibility genes (Fig 4b). Is this due to more patient samples from these tumor types, or an intrinsic biological difference?

Response: We thank the reviewer for the great suggestion. We have conducted a new analysis and found no significant correlation between the sample size of these tumor types and the number of significant 3'aTWAS genes (**New Figure S16a**). Utilizing the estimated heritability data from Table S2, we further assessed the correlation between heritability and the number of significant 3'aTWAS genes. We found that there is a strong correlation between heritability and the number of significant 3'aTWAS genes ($R = 0.93$, $P = 1.2 \times 10^{-12}$, **New Figure S16b**), suggesting that the greatest number of APA-linked susceptibility genes may be due to the high heritability of these two cancer types, rather than sample size differences.

New Figure S16. The correlations between sample sizes, heritability and the number of identified APA-linked genes across cancer types. **a.** The GWAS sample size shows no significant correlation with the number of identified significant 3'aTWS genes (FDR<0.05). **b.** There is a strong correlation between estimated heritability and the number of significant 3'aTWS genes (FDR<0.05). The coefficient (R) and P-value were calculated from Pearson correlation.

2) In Fig 4, the authors claim that APA-linked susceptibility genes tend to have greater impact on cancer cell proliferation, relative to non-3'aTWS genes. However, is this the case when compared to splicing and expression TWAS genes?

Response: We thank the reviewer for the suggestion. We have now compared our APA-linked susceptibility genes with splicing and expression TWAS genes (**New Figure. S18**), we found the mean CERES scores of our APA-linked susceptibility genes have a significant impact on cancer cell proliferation than expression or splicing. We have thus revised this in the revised manuscript.

New Figure S18. Comparisons of the mean CERES score among significant 3'aTWAS genes versus expression TWAS and splicing TWAS. The *P*-value was calculated from the Wilcoxon's test.

3) It is unclear how the 3'RACE experiment was performed.

Response: We apologize for the confusing explanation. We have now revised the sentences in the revised manuscript for better clarity: *"To begin with, the previously described luciferase psiCHECK-2 vector (constructed by Tsingke biotech and confirmed by sequencing) were transfected into cells using Lipofectamine 3000 Transfection Reagent (Invitrogen, cat#: L3000015). After 48 hours, cells were harvested for RNA extraction. Total RNA was extracted using TRIzol reagent according to the manufacturer's instruction. Next, The the full length of 3'UTR was identified and amplified from the total RNA of transfected MCF-7 cells by 3'RACE using the GoScript™ Reverse Transcriptase (Promega, cat#: A5001) with oligo dT18-XbaKpnBam primer following the manufacturer's protocol. Finally, 3'RACE was carried out using the check-luc forward primer and XbaKpnBam reverse primer to distinguish exogenous RNA from endogenous RNA."*

-Was this detecting exogenous DNA?

Response: Yes, we detected the exogenous DNA following the protocol described previously⁴.

-Please draw arrows next to the gel highlighting the expected short and long forms (Fig 5e).

Response: We thank the suggestion and have now updated figures to highlight the expected short and long forms.

New Figure 5e. Changes of 3'UTR usage in MCF-7 cells detected by 3' RACE. The 3' RACE strategy is shown in the upper panel. F and R indicate forward and reverse primers, respectively. The bar plot on the right shows the usage of dPAS measured through image lab.

4) While the authors clearly demonstrate that the long form of CRLS1 3'UTR drives increased protein expression, and that CRLS1 mediates breast cancer cell growth (to a modest extent), there is no evidence that the specific variant contributes to cell proliferation (line 344). To support this conclusion, the authors could overexpress both the WT and variant forms in non-transformed mammary epithelial cells and measure cell proliferation. Or the authors could alter their conclusions.

Response: We thank the reviewer's valuable suggestion. We agree with the reviewer and have thus updated the discussion in our revised manuscript according. "Our findings demonstrated that the cancer-associated noncoding variant impacts 3'UTR usage, resulting in altered protein abundance which ultimately contributes to cell proliferation."

5) On lines 333-335, the authors make a bold claim that these variants play a significant role in cancer initiation. While this may be true, this is not tested experimentally in their manuscript. I would suggest toning down this statement.

Response: We thank the reviewer for the suggestion. We agree with the reviewer and

have thus toned down the statement as follows “Furthermore, our findings provide additional evidence that common genetic variants linked to alternative 3’UTR usage can contribute to the cancer risk and may play a substantial role in cancer initiation”.

Reference

1. Lindstrom, S. *et al.* Genome-wide association study identifies multiple loci associated with both mammographic density and breast cancer risk. *Nat Commun* **5**, 5303 (2014).
2. Dhanasekaran, R. *et al.* The MYC oncogene - the grand orchestrator of cancer growth and immune evasion. *Nat Rev Clin Oncol* **19**, 23-36 (2022).
3. Yao, C. *et al.* Transcriptome-wide analyses of CstF64-RNA interactions in global regulation of mRNA alternative polyadenylation. *Proc Natl Acad Sci U S A* **109**, 18773-8 (2012).
4. Zhao, Z. *et al.* Comprehensive characterization of somatic variants associated with intronic polyadenylation in human cancers. *Nucleic Acids Res* **49**, 10369-10381 (2021).
5. Tang, H.W. *et al.* The TORC1-Regulated CPA Complex Rewires an RNA Processing Network to Drive Autophagy and Metabolic Reprogramming. *Cell Metab* **27**, 1040-1054 e8 (2018).
6. Zhao, T. *et al.* Transcriptomics-proteomics Integration reveals alternative polyadenylation driving inflammation-related protein translation in patients with diabetic nephropathy. *J Transl Med* **21**, 86 (2023).
7. Chen, H.F., Hsu, C.M. & Huang, Y.S. CPEB2-dependent translation of long 3'-UTR Ucp1 mRNA promotes thermogenesis in brown adipose tissue. *EMBO J* **37**(2018).
8. Rieger, M.A. *et al.* CLIP and Massively Parallel Functional Analysis of CELF6 Reveal a Role in Destabilizing Synaptic Gene mRNAs through Interaction with 3' UTR Elements. *Cell Rep* **33**, 108531 (2020).

REVIEWERS' COMMENTS

Reviewer #1 (Remarks to the Author):

The reviewers addressed all my comments.

Reviewer #2 (Remarks to the Author):

My concerns are fully addressed.

Reviewer #3 (Remarks to the Author):

The authors addressed all previous critiques. Really nice work overall!